# Rebalancing of actomyosin contractility enables mammary tumor formation upon loss of E-cadherin

Koen Schipper[1], Danielle Seinstra[1], Anne Paulien Drenth[1], Eline van der Burg[1], Veronika Ramovs[2], Arnoud Sonnenberg[2], Jacco van Rheenen[1], Micha Nethe[1,3] & Jos Jonkers[1]

E-cadherin (*CDH1*) is a master regulator of epithelial cell adherence junctions and a well-established tumor suppressor in Invasive Lobular Carcinoma (ILC). Intriguingly, somatic inactivation of E-cadherin alone in mouse mammary epithelial cells (MMECs) is insufficient to induce tumor formation. Here we show that E-cadherin loss induces extrusion of luminal MMECs to the basal lamina. Remarkably, E-cadherin-deficient MMECs can breach the basal lamina but do not disseminate into the surrounding fat pad. Basal lamina components laminin and collagen IV supported adhesion and survival of E-cadherin-deficient MMECs while collagen I, the principle component of the mammary stromal micro-environment did not. We uncovered that relaxation of actomyosin contractility mediates adhesion and survival of E-cadherin-deficient MMECs on collagen I, thereby allowing ILC development. Together, these findings unmask the direct consequences of E-cadherin inactivation in the mammary gland and identify aberrant actomyosin contractility as a critical barrier to ILC formation.

[1] Division of Molecular Pathology, Oncode Institute, The Netherlands Cancer Institute, Amsterdam, The Netherlands. [2] Division of Cell Biology, The Netherlands Cancer Institute, Amsterdam, The Netherlands. [3] Present address: Department of Hematopoiesis, Sanquin Research and Landsteiner Laboratory, Academic Medical Center, Amsterdam, The Netherlands. Correspondence and requests for materials should be addressed to M.N. (email: m.nethe@sanquin.nl) or to J.J. (email: j.jonkers@nki.nl)

The maintenance of epithelial cell–cell adhesion sustains organ function and prevents tissue infection and tumor development. E-cadherin (*CDH1*) is a transmembrane protein and a master regulator of the cell adherence junctions controlling cell–cell adhesion. E-cadherin is also a well-established tumor suppressor which is in general viewed to prevent tumor cell dissemination[1–3]. Complete loss of E-cadherin expression is a hallmark of invasive lobular carcinoma (ILC)[4]. ILC comprises 8–14% of all breast cancer patients, making it the 2nd most diagnosed breast cancer type worldwide[5,6]. Histologically, ILC is characterized by non-cohesive cells arranged in single files infiltrating the surrounding fibrous collagen-rich stroma[7]. The infiltrative nature of ILC often complicates surgical removal. ILCs also typically respond poorly to classical regimes of chemotherapy. To gain insight into the mechanisms underlying ILC development, we have previously developed genetically engineered mouse models that, closely resemble human ILC[8,9]. Mammary gland-specific inactivation of E-cadherin in combination with loss of p53 or PTEN markedly enhanced mammary tumor development thus confirming the tumor suppressive role of E-cadherin[8,9]. Intriguingly, somatic inactivation of E-cadherin by itself is not sufficient to induce mammary tumor formation, but rather seems to hamper survival of mouse mammary epithelial cells (MMECs)[8,10]. In addition, genetic inactivation of E-cadherin in MMEC lines or mouse mammary epithelial organoids does not induce single cell dissociation nor epithelial-mesenchymal transition (EMT)[11,12]. It remains unclear why loss of E-cadherin by itself hampers cell survival and how this limitation is circumvented during ILC development.

Here we studied the direct consequence of E-cadherin loss in MMECs by genetic labeling and tracing of E-cadherin-deficient MMECs in vivo. We found that E-cadherin-deficient MMECs extrude into the fibrous stroma where they form dynamic but non-tumorigenic cell clusters that only progress to ILC upon partial relaxation of actomyosin. These findings show that actomyosin levels are critical in ILC development.

## Results

**E-cadherin loss drives cell extrusion to the basal lamina.** To monitor the fate of E-cadherin-deficient MMECs in vivo, we generated *Wcre;Cdh1^{F/F};mTmG* mice in which mammary gland-specific expression of Cre from the *Wap* gene promoter (*Wcre*) induces somatic inactivation of *Cdh1* (encoding E-cadherin) and GFP expression by recombination of the mTmG reporter in MMECs[13,14] (Fig. 1a). Cre recombinase activity, marked by GFP expression, was exclusively detected in cytokeratin-8 (CK8)-positive luminal MMECs (Fig. 1b, c). Strikingly, immunofluorescence (IF) analysis of GFP expression in 6-week-old *Wcre;Cdh1^{F/F};mTmG* mice revealed that E-cadherin-deficient luminal MMECs massively extruded towards the basal lamina and typically resided between the layer of cytokeratin-14 (CK14)-positive myoepithelial cells and the basal stroma (Fig. 1d). Loss of functional E-cadherin in MMECs was also confirmed by dissociation of both β-catenin and p120-catenin from the peripheral membrane due to disruption of the E-cadherin–catenin complex as previously described (Supplementary Fig. 1a, b)[12,15]. Whereas most extruded luminal MMECs were detected at the basal laminal border, apoptotic E-cadherin-deficient MMECs were sporadically detected in the lumen of the mammary ducts as observed previously[8,10]. To monitor the fate of extruded E-cadherin-deficient MMECs, we compared mammary gland sections of 3-, 5-, and 12-month-old *Wcre;Cdh1^{F/F};mTmG* mice (*n* = 3) to age-matched *Wcre;mTmG* mice by immunohistochemistry (IHC) (Fig. 1e). Interestingly, extruded GFP-marked E-cadherin-deficient MMECs in mammary glands of *Wcre;Cdh1^{F/F};mTmG*

mice accumulated as small clusters of cells in the fibrous surrounding stroma. IF analysis confirmed lack of E-cadherin expression in extruded GFP-positive MMECs (Fig. 1f). Moreover, the extruded MMECs represented the majority of GFP-marked E-cadherin-deficient MMECs in *Wcre;Cdh1^{F/F};mTmG* mice, whereas no extrusion of GFP-positive control MMECs was observed in *Wcre;mTmG* mice (Fig. 1e, g). The clusters of extruded cells did not increase in size over time, which is in line with our previous observation that loss of E-cadherin by itself does not induce mammary tumor formation in mice (Fig. 1h)[8]. Finally, we did not detect any MMECs in the lumen of mammary glands at these time points, supporting previous findings that E-cadherin-deficient MMECs that extrude into the lumen of the mammary gland undergo apoptosis and are rapidly cleared[8,10].

**E-cadherin loss in MMECs increases actomyosin contractility.** To gain more insight into the behavior of the extruded E-cadherin-deficient MMECs in the basal mammary stroma in situ, we analyzed mammary glands of 3-month old *Wcre;Cdh1^{F/F}; mTmG* mice and *Wcre;mTmG* control mice (*n* = 3) by live intravital microscopy. The E-cadherin-deficient cellular clusters in *Wcre;Cdh1^{F/F};mTmG* mice were present alongside the entire mammary ductal tree (visualized by mTomato) and faced the surrounding mammary stroma (Fig. 2a). GFP-marked extruded E-cadherin-deficient MMECs formed tight but highly dynamic clusters of motile cells which appeared to constantly tumble around each other (Fig. 2b, Supplementary Movies 1–3). Despite their enhanced motility within these clusters, E-cadherin-deficient cells did not disseminate into the surrounding mammary stroma. Interestingly, extruded MMECs in mammary glands of *Wcre;Cdh1^{F/F};mTmG* mice were marked by extensive membrane blebbing (Fig. 2c, d). Membrane blebbing is often seen in amoeboid migration[16] and apoptosis[17]. However we could not observe any defined form of cell movement or cell death during the time of imaging. We also did not find any cleaved caspase-3-positive apoptotic cells at the basal stromal compartment[8]. Since membrane blebbing typically results from elevated actomyosin contractility, we next examined myosin light chain (MLC) phosphorylation by IF imaging in mammary gland sections of *Wcre;mTmG* and *Wcre;Cdh1^{F/F};mTmG* mice (Fig. 2e). In normal mammary glands, luminal epithelial cells have relatively low MLC phosphorylation levels compared to myoepithelial cells. E-cadherin-deficient MMECs in the mammary fibrous stroma showed a clear increase in pMLC staining, confirming an increase in actomyosin contractility (Fig. 2e, f, Supplementary Fig. 2a). Overall these results reveal that E-cadherin-deficient MMECs that persist in the fibrous mammary stroma exhibit an increase in actomyosin contractility.

**Actomyosin relaxation enables survival upon E-cadherin loss.** To study the consequences of E-cadherin loss in more detail, we isolated mammary epithelium from *Wcre;Cdh1^{F/F};mTmG* mice and *Wcre;mTmG* mice. Initial attempts to culture *Wcre;Cdh1^{F/F}; mTmG* MMECs failed as GFP-positive MMECs detached during formation of epithelial islands and were consequently lost during cell culture (Fig. 3a, Supplementary Movie 4). Since extensive cell blebbing preceded cell detachment, we hypothesized that increased actomyosin contractility, as observed in situ, may hamper adhesion of *Wcre;Cdh1^{F/F};mTmG* MMECs. To test this hypothesis we cultured *Wcre;Cdh1^{F/F};mTmG* MMECs with a ROCK inhibitor (Y-27632), which rescued their adhesion and expansion (Fig. 3a). The obtained polyclonal MMEC population, harboring both GFP-positive and GFP-negative MMECS, had decreased E-cadherin expression indicating the presence of E-cadherin-negative cells (Fig. 3b). Next, we derived E-cadherin-

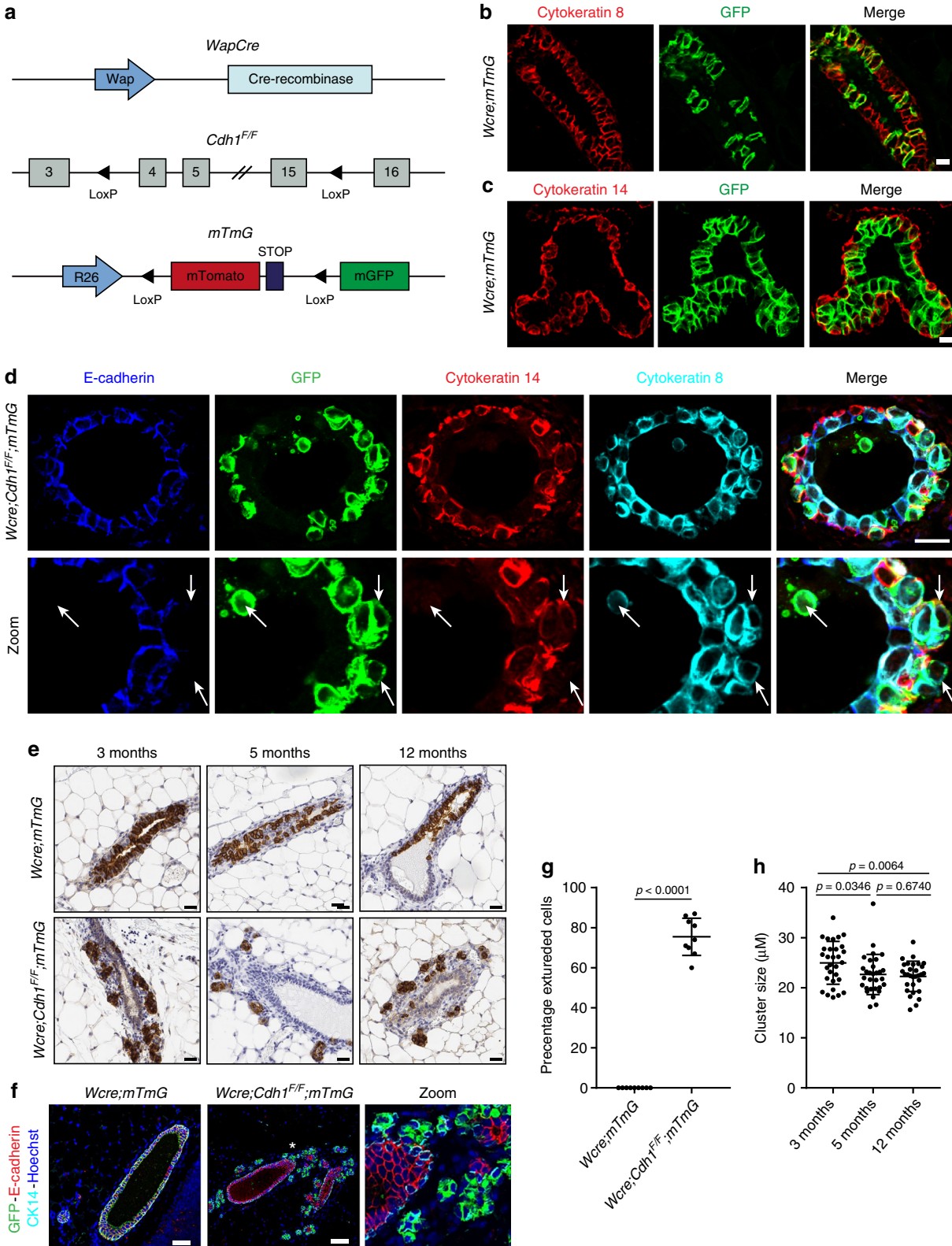

proficient and -deficient subclones by picking GFP-positive MMECS in the presence of Y-27632 from *Wcre;mTmG* and *Wcre;Cdh1^{F/F};mTmG* MMECs (Fig. 3c). This allowed us to examine cell adhesion of cultured E-cadherin-deficient *Wcre; Cdh1^{F/F};mTmG* MMEC subclones in the presence and absence of Y-27632 (10 µM). All E-cadherin-deficient MMEC subclones displayed extensive cell blebbing during cell adhesion (Fig. 3d). In

the absence of Y-27632, E-cadherin-deficient MMECs showed reduced survival signaling and increased expression of cleaved caspase-3 indicating the initiation of apoptosis (Fig. 3d). Inhibition of ROCK activity by Y-27632 rescued cell spreading and survival of *Wcre;Cdh1^{F/F};mTmG* MMECs (Fig. 3d). To reduce actomyosin contractility downstream of ROCK we next used Blebbistatin to directly reduce myosin activity. Like ROCK

**Fig. 1** E-cadherin loss drives cell extrusion towards the basal lamina. **a** Schematic overview of engineered alleles in *Wcre;Cdh1^F/F^;mTmG* mice.
**b**, **c** Examination of GFP-positive Wcre activity in mammary glands of 6-week-old *Wcre;mTmG* female mice by immunofluorescence (IF) analysis (*n* = 6). **b** IF staining of GFP was examined in cytokeratin-8 (CK8)-positive luminal mouse mammary epithelial cells (MMECs) and **c** cytokeratin-14 (CK14)-positive myoepithelial cells. Scale bar, 10 µm. **d** Identification of E-cadherin inactivated MMECs by IF staining of E-cadherin, GFP, CK14, and CK8 in 6-week-old *Wcre;Cdh1^F/F^;mTmG* female mice (*n* = 6). Arrows indicate events of cell extrusion upon inactivation of E-cadherin. Scale bare, 10 µm. **e** Immunohistochemical detection of GFP-positive E-cadherin inactivated MMECs in mammary gland sections of 3-, 4-, and 6-month-old *Wcre;Cdh1^F/F^;mTmG* female mice and age-matched *WCre;mTmG* control mice (*n* = 3). Scale bar, 20 µm. **f** Examination of E-cadherin expression in mammary gland sections of 3-month-old *Wcre;Cdh1^F/F^;mTmG* female mice and age-matched *Wcre;mTmG* control mice by IF analysis of GFP, E-cadherin, CK14, and Hoechst. Asterisk indicates area of zoom. Scale bar, 50 µm. **g** Quantification of the amount of extruded GFP-positive cells in 3-month-old *Wcre;Cdh1^F/F^;mTmG* (*n* = 3) female mice and age-matched *Wcre;mTmG* (*n* = 3) control mice. Data are of nine images per group. **h** Quantification of the average size of extruded GFP-positive cell clusters in the mammary glands of *Wcre;Cdh1^F/F^;mTmG* mice at the ages of 3, 5, and 12 months. Data are of three mice per time point and 10 images per mouse. All data are depicted as mean ± standard deviation. All *p* values were calculated using an unpaired two tailed *t*-test. Source data are provided as a Source Data file

inhibition by Y-27632, direct inhibition of myosin activity by blebbistatin (10 µM) readily rescued cell adhesion of E-cadherin-deficient *Wcre;Cdh1^F/F^;mTmG* MMECs (Fig. 3d). E-cadherin-proficient *Wcre;mTmG* MMEC subclones did not require Y-27632 or blebbistatin to induce cell spreading but nevertheless displayed enhanced cell adhesion and survival upon inhibiting actomyosin contractility (Fig. 3d, f). Remarkably, once cell adhesion was restored, *Wcre;Cdh1^F/F^;mTmG* MMECs grew significantly faster than *Wcre;mTmG* MMECs, likely due to loss of cell contact inhibition as previously reported to be controlled by E-cadherin signaling (Fig. 3f, g)[18,19]. To substantiate our findings, we derived polyclonal cell lines from wild-type (WT) MMECs and *Cdh1^Δ/Δ^* MMECs, which were produced by exposing *Cdh1^F/F^* MMECs to a Cre-encoding adenovirus (Fig. 3h). The *Cdh1^Δ/Δ^* MMEC line showed a similar dependency on actomyosin relaxation as the *Wcre;Cdh1^F/F^;mTmG* MMEC subclones (Fig. 3i, j). Previously, E-cadherin has been shown to restrict RhoA activity by controlling both p190RhoGAP and p120-catenin signaling[20–22]. To determine if RhoA activity was also increased in E-cadherin-deficient MMECs during cell adhesion, we performed a RhoA activation pull-down assay. Indeed, *Cdh1^Δ/Δ^* MMECs had increased levels of GTP-bound RhoA which coincided with increased expression of total RhoA compared to WT MMECs (Fig. 3k, i, Supplementary Fig. 3a). *Cdh1^Δ/Δ^* MMECs also showed increased MLC expression and phosphorylation, confirming an overall increase in actomyosin contractility upon cell adhesion compared to WT MMECs. (Fig. 3k, l, Supplementary Fig. 3a–c). Together these results show that E-cadherin-deficient MMECs have aberrant actomyosin contractility during cell adhesion which needs to be reduced in order to allow their adhesion and survival.

**Balanced actomyosin contractility is essential for survival.**
Whereas E-cadherin-deficient MMECs require actomyosin relaxation in order to sustain cell adhesion and survival, progression of cell division requires actomyosin contraction[23,24]. We therefore hypothesized that too much actomyosin relaxation would hamper cell survival of E-cadherin-deficient MMECs. To examine this in more detail, we analyzed survival of E-cadherin-deficient MMECs subjected to increasing levels of actomyosin relaxation. We performed colony formation assays using concentration ranges of Blebbistatin (Fig. 4a, b) or the ROCK inhibitor H1152 (ref. [25]) (Fig. 4c, d). H1152 inhibits ROCK with increased specificity and potency compared to Y-27632. Both inhibitors readily rescued survival of E-cadherin-deficient MMECs at low concentrations, i.e. 0.5 µM for Blebbistatin and 0.1 µM for H1152 (Fig. 4a–d). Optimal concentrations permitting survival of E-cadherin-deficient MMECs were found at 5 µM for Blebbistatin and 1 µM for H1152. Importantly, higher concentrations of both

compounds hampered survival of E-cadherin-deficient MMECs. Exposure of E-cadherin-deficient MMECs to various concentrations of H1152 reveals a dose-dependent effect on MLC phosphorylation (Supplementary Fig. 4). The optimal concentration of H1152 only partially inhibited MLC phosphorylation indicating that too much relaxation is not tolerated. Altogether these results underscore the requirement of a critical balance in actomyosin contractility that mediates adhesion but does not hamper survival of E-cadherin-deficient MECs.

**MYPT1 drives actomyosin relaxation and cell adhesion**. We recently found that protein-truncating mutations in *Ppp1r12a* (encoding MYPT1) drive ILC formation in mice[26]. MYPT1 regulates actomyosin contractility by acting as the regulatory subunit of the holoenzyme myosin light chain phosphatase MLCP which inhibits myosin light chain (MLC) activity[27]. Truncated MYPT1 lacks the PLK1 and LATS1 interaction domains involved in regulation of mitosis and retains the functional domains to bind PP1 and target pMLC[28–30]. Truncated MYPT1 also lacks the two inhibitory phosphorylation sites (T696 and T853) that are targeted by ROCK and cause auto-inhibition of MYPT1 (ref. [29]). Therefore, we wondered if truncated MYPT1 could inhibit actomyosin contractility and thereby rescue adhesion of *Wcre;Cdh1^F/F^;mTmG* MMECs upon Y-27632 withdrawal. To test this hypothesis, we transduced the E-cadherin-deficient *Wcre;Cdh1^F/F^;mTmG* MMEC subclones with a lentivirus encoding doxycycline (dox)-inducible, Flag-tagged truncated MYPT1 (t-MYPT1) (Fig. 5a). Protein expression of t-MYPT1 was clearly detected 24 h after adding dox and lost 48 h after dox washout (Fig. 5b). Loss of t-MYPT1 significantly increased MLC activity, reflected by increased pMLC IF staining along the actin stress fibers confirming that t-MYPT1 reduces actomyosin contractility (Fig. 5c, d). In accordance with exposure to optimal concentrations of H1152 expression of tMYPT1 does not completely inhibit MLC phosphorylation in E-cadherin-deficient MMECs. Accordingly, the expression of t-MYPT1 also rescued cell adhesion and survival of *Wcre;Cdh1^F/F^;mTmG* MMECs in the absence of Y-27632 (Fig. 5e, f). We further hypothesized that the absence of the ROCK-controlled inhibitory phosphorylation sites would increase the potency of t-MYPT1 as an inhibitor of actomyosin contractility (Fig. 5g). Indeed, overexpression of full-length MYPT1 had a relatively modest effect on MLC phosphorylation and cell adhesion compared to t-MYPT1 (Fig. 5h–i, Supplementary Fig. 4). To reduce actomyosin contractility, MYPT1 encompasses a N-terminal region containing a protein phosphatase 1 (PP1) recognition motif and seven Ankyrin repeats (ANK) that facilitate PP1 binding, all of which are retained in t-MYPT1[27,31,32]. PP1 acts as the catalytic subunit of MLCP and reduces actomyosin contractility by dephosphorylating Serine 19 of MLC[32]. Since the

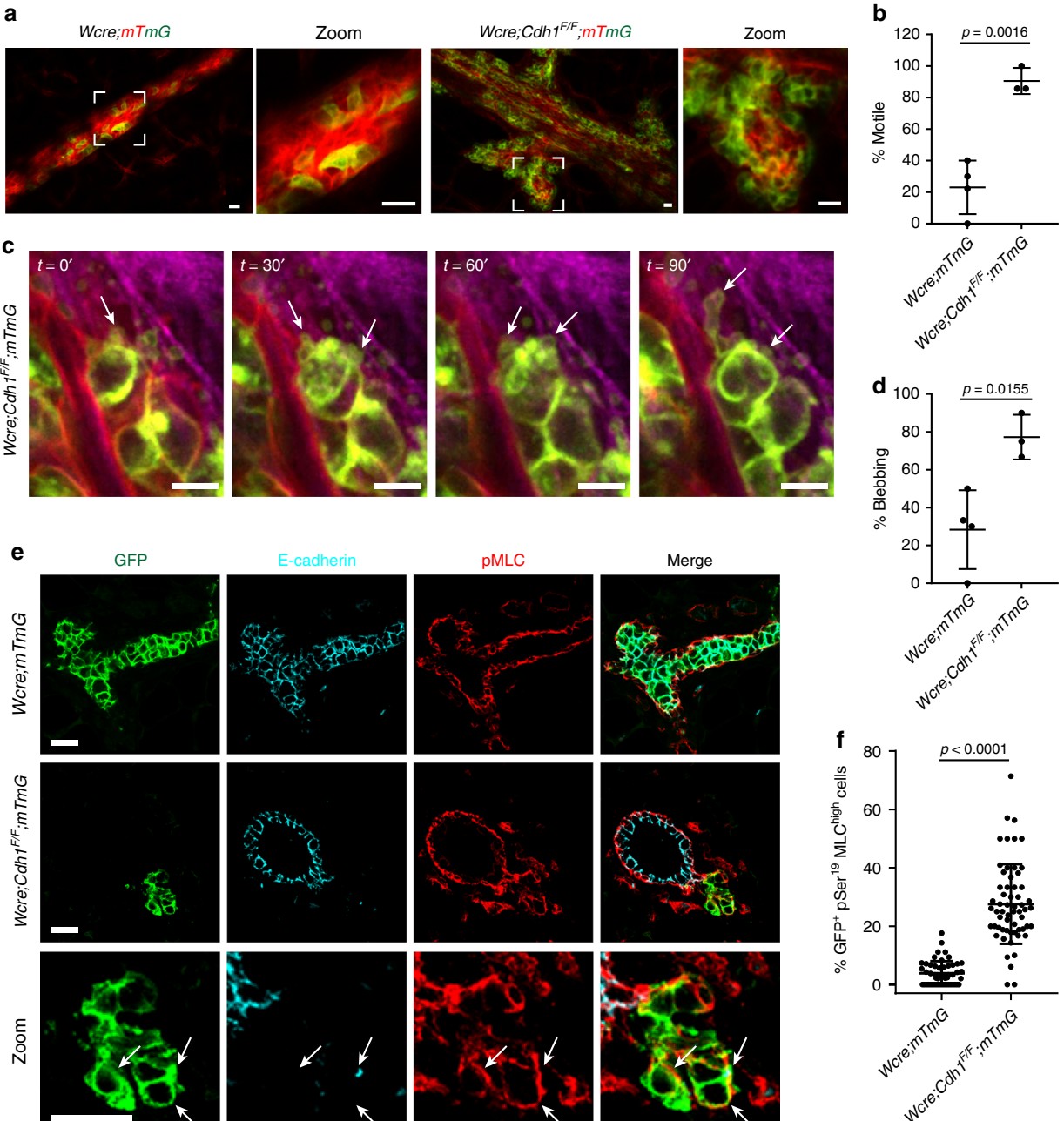

**Fig. 2** E-cadherin loss increases actomyosin contractility. **a** Still images derived from in vivo intravital imaging of the mammary gland of 8-week-old *Wcre; mTmG* and *Wcre;Cdh1^{F/F};mTmG* mice displaying GFP-positive Cre-switched MMECs and mTomato non-switched MMECs and stromal cells. Zooms reveal motile GFP-positive E-cadherin inactivated MMECs in *Wcre;Cdh1^{F/F};mTmG* mice. Scale bars, 20 μm. **b** Quantification of the percentage of GFP-positive motile cells among *Wcre;mTmG* ($n = 2$) and *Wcre;Cdh1^{F/F};mTmG* ($n = 3$) mice. Data are of four and three images per group, respectively. **c** Still image of in vivo intravital imaging of *Wcre;Cdh1^{F/F};mTmG* mice demonstrates extensive cell blebbing of GFP-positive E-cadherin inactivated MMECs. Scale bar, 10 μm. **d** Quantification of the percentage of GFP-positive blebbing cells among *Wcre;mTmG* ($n = 2$) and *Wcre;Cdh1^{F/F};mTmG* ($n = 3$) mice. Data are of four and three images per group, respectively. **e** Examination of myosin light chain (MLC) activity by IF analysis of pSer19 MLC in GFP-positive E-cadherin-deficient MMECs in 5-month-old *Wcre;mTmG* and *Wcre;Cdh1^{F/F};mTmG* mice. Scale bar, 20 μm. **f** Quantification of the amount of GFP^+ pSer19 MLC^high cells *Wcre;mTmG* ($n = 3$) and *Wcre;Cdh1^{F/F};mTmG* ($n = 3$) mice. Data are of 51 and 60 images, respectively. All data are depicted as mean ± standard deviation. All p values were calculated using an unpaired two tailed *t*-test. Source data are provided as a Source Data file

PP1 recognition motif of MYPT1 has been shown to drive PP1 activity upon binding of PP1[32], we generated a t-MYPT1 variant lacking the PP1 recognition motif (t-MYPT1ΔPP1) (Fig. 5g). Stable expression of t-MYPT1 and t-MYPT1ΔPP1 showed that loss of the PP1 recognition domain impaired dephosphorylation of pMLC in *Wcre;Cdh1^{F/F};mTmG* MMECs (Fig. 5h, i). Expression

of t-MYPT1ΔPP1 also failed to rescue cell adhesion and survival of *Wcre;Cdh1^{F/F};mTmG* MMECs in clonogenic survival assays (Fig. 5j, k, Supplementary Fig. 5). Overall, these data demonstrate that t-MYPT1 rescues cell adhesion and survival of *Wcre;Cdh1^{F/F}; mTmG* MMECs by PP1-mediated relaxation of actomyosin contractility.

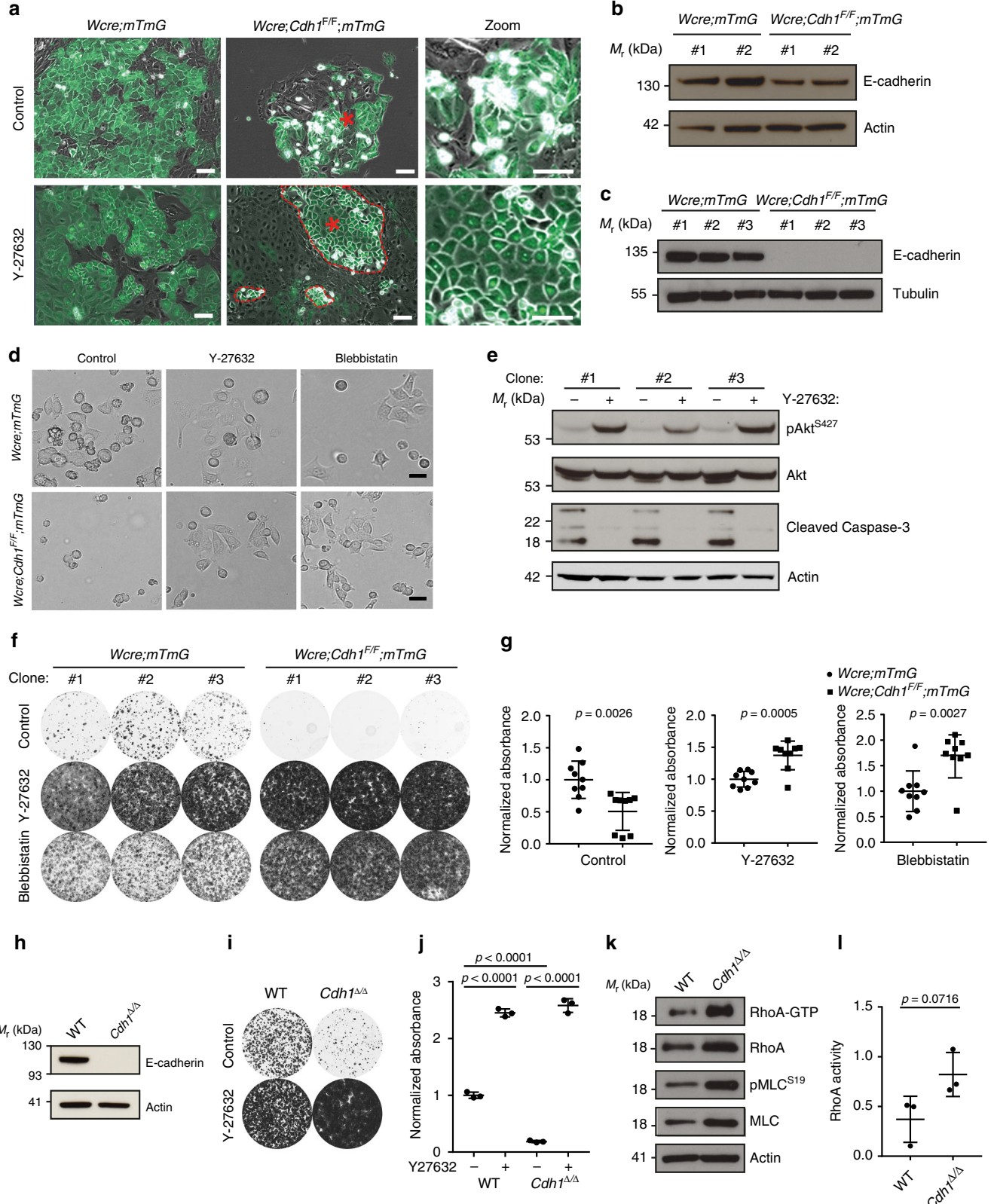

**Collagen I does not support adhesion upon loss of E-cadherin.** Since loss of E-cadherin impaired cell adhesion of primary MMECs in vitro, we next examined the impact of the extracellular matrix surrounding the E-cadherin-deficient MMECs. Upon Cre expression in 6-week-old *Wcre;Cdh1^{F/F};mTmG* mice, GFP-marked E-cadherin-deficient MMECs extruded towards the basal lamina (Fig. 6a, zoom). These findings are in line with the observed extruded clusters of E-cadherin-deficient MMECs in the surrounding mammary fibrous stroma which we started to observe in 12 weeks and older *Wcre;Cdh1^{F/F};mTmG* mice (Fig. 1e, f). Remarkably, these clusters in the mammary fibrous stroma are surrounded by laminin (Fig. 6a). We therefore next examined cell

**Fig. 3** Actomyosin relaxation enables survival upon E-cadherin loss. **a** Merged brightfield and GFP images of MMECs isolated from *Wcre;mTmG* and *Wcre; Cdh1$^{F/F}$;mTmG* female mice in the absence or presence of 10 μM Y-27632. Asterisks indicate areas of zoom, red dotted lines display outgrowth of E-cadherin-deficient MMECs. Zooms display altered cell adhesion in the absence or presence of Y-27632 of GFP-positive E-cadherin-deficient cells derived from *Wcre;Cdh1$^{F/F}$;mTmG* female mice. Scale bar, 50 μm. **b** Western blot analysis of *Wcre;mTmG* and *Wcre;Cdh1$^{F/F}$;mTmG* MMECs stained for E-cadherin and tubulin (loading control). **c** Western blot analysis of subclones derived from the *Wcre;mTmG* and *Wcre;Cdh1$^{F/F}$;mTmG* MMECs stained for E-cadherin and tubulin (loading control). **d** Brightfield images of MMECs 3 h post seeding in the absence or presence of 10 μM Y-27632 or 10 μM blebbistatin. Scale bar, 20 μm. **e** Western blot analysis of pSer$^{473}$ Akt, Akt, cleaved caspase-3, and actin (loading control) in *Wcre;Cdh1$^{F/F}$;mTmG* subclones grown in the absence or presence of 10 μM Y-27632. **f, g** Representative images (**f**) and quantification (**g**) of clonogenic assays with *Wcre;mTmG* (circles) and *Wcre; Cdh1$^{F/F}$;mTmG* (squares) subclones grown in the presence or absence of 10 μM Y-27632 or 10 μM blebbistatin (7 days after seeding the cells). Data are of three independent experiments with three clones per experiment. **h** Western blot analysis of WT FVB and *Cdh1$^{Δ/Δ}$* MMECs stained for E-cadherin and actin (loading control). **i, j** Representative images (**i**) and quantification (**j**) of clonogenic assays with *Cdh1$^{Δ/Δ}$* and WT control MMEC lines with or without 10 μM Y-27632. Data are of three independent experiments. **k** Representative images of immunoblots of active RhoA pull-down assays and ser$^{19}$ myosin light chain phosphorylation (pMLC) from *Cdh1$^{Δ/Δ}$* and WT control MMECs harvested 3 h post seeding. **l** Quantification of active RhoA pull-down assays by densitometry normalized to the actin loading control. Data are of three independent experiments. All data are depicted as mean ± standard deviation. All *p* values were calculated using an unpaired two tailed *t* test. Source data are provided as a Source Data file

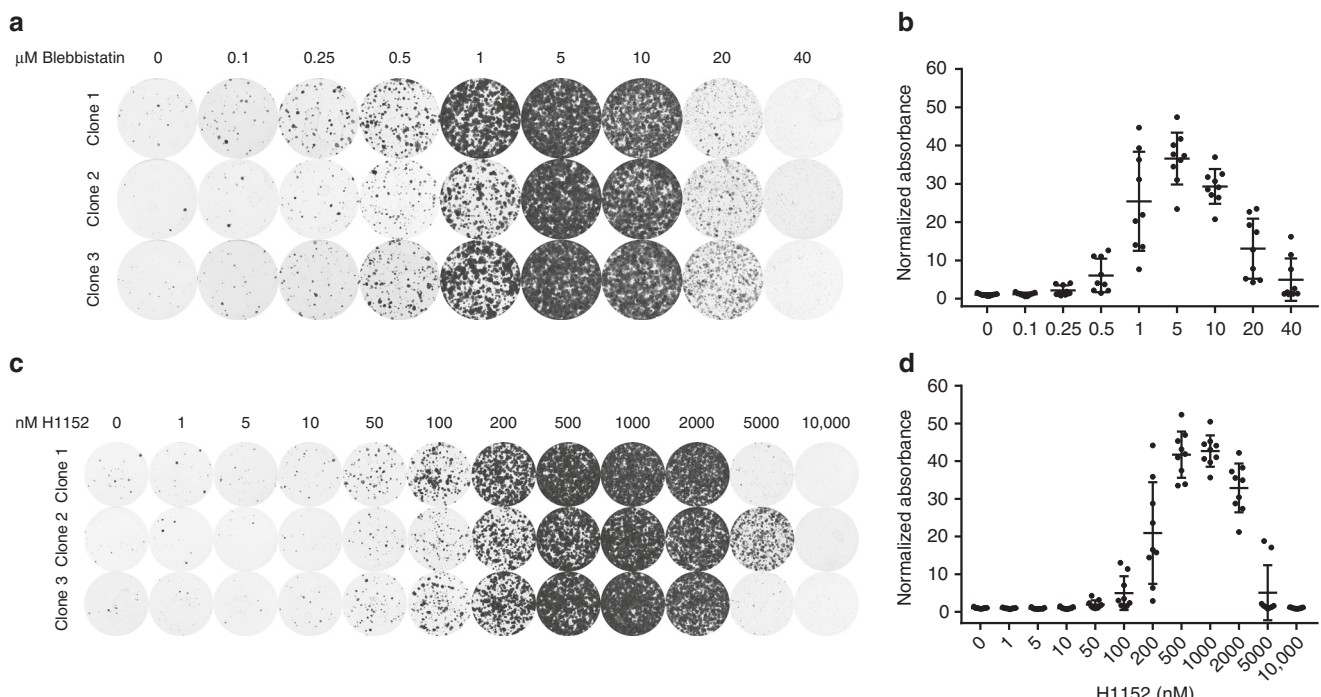

**Fig. 4** Balanced actomyosin contractility is essential for survival. **a, b** Representative images (**a**) and quantification (**b**) of clonogenic assays with *Wcre; Cdh1$^{F/F}$;mTmG* MMECs cultured with different concentrations of blebbistatin. Data are of three independent experiments with three clones per experiment. **c, d** Representative images (**c**) and quantification (**d**) of clonogenic assays with *Wcre;Cdh1$^{F/F}$;mTmG* MMECs cultured with different concentrations of H1152. Data are of three independent experiments with three clones per experiment. All data are depicted as mean ± standard deviation. Source data are provided as a Source Data file

adhesion and survival of E-cadherin-deficient MMECs on laminin-332, a component of the mammary basal lamina. Laminin-332-rich matrix derived from Rac11P cells readily supported cell adhesion and survival of *Wcre;Cdh1$^{F/F}$;mTmG* MMECs in the absence of Y-27632 or Blebbistatin and therefore likely supports survival of E-cadherin-deficient MMECs in the fibrous stroma (Fig. 5b, c). In comparison to laminin-332, collagen IV, another component of the basal lamina, induced a more modest increase in survival of E-cadherin-deficient MMECs (Supplementary Fig. 6). Yet, the mammary fibrous stroma is mainly characterized by the deposition of collagen I[33]. The clusters of E-cadherin-deficient MMECs, marked by GFP, in the mammary glands of *Wcre;Cdh1$^{F/F}$;mTmG* mice were indeed surrounded by fibrous collagen (Fig. 6d, Supplementary Fig. 7). We therefore compared cell adhesion and survival of E-cadherin-deficient MMECs on

laminin-322 with collagen l (Fig. 6e, f). In contrast to laminin-332, collagen l did not support adhesion and survival of E-cadherin-deficient MMECs despite expression of the collagen-receptor integrin α2β1 (Fig. 6e, f, Supplementary Fig. 8). Contrary to laminin-332, adhesion to collagen I has been reported to result in increased RhoA activity[34,35]. Indeed, a RhoA activation assay confirmed that *Cdh1$^{Δ/Δ}$* MMECs on laminin-332 matrix have significantly decreased RhoA activity compared to cells on collagen I during cell adhesion (Fig. 6g–i). In line with these findings, survival of E-cadherin-deficient MMECs on collagen I could be rescued by expression of t-MYPT1 but not by t-MYPT1ΔPP1, which is unable to reduce actomyosin contractility (Fig. 6j, k). Overall these results highlight that collagen l impairs cell adhesion and survival of E-cadherin-deficient MMECs, indicating that the collagen-rich mammary stroma may play an important role in

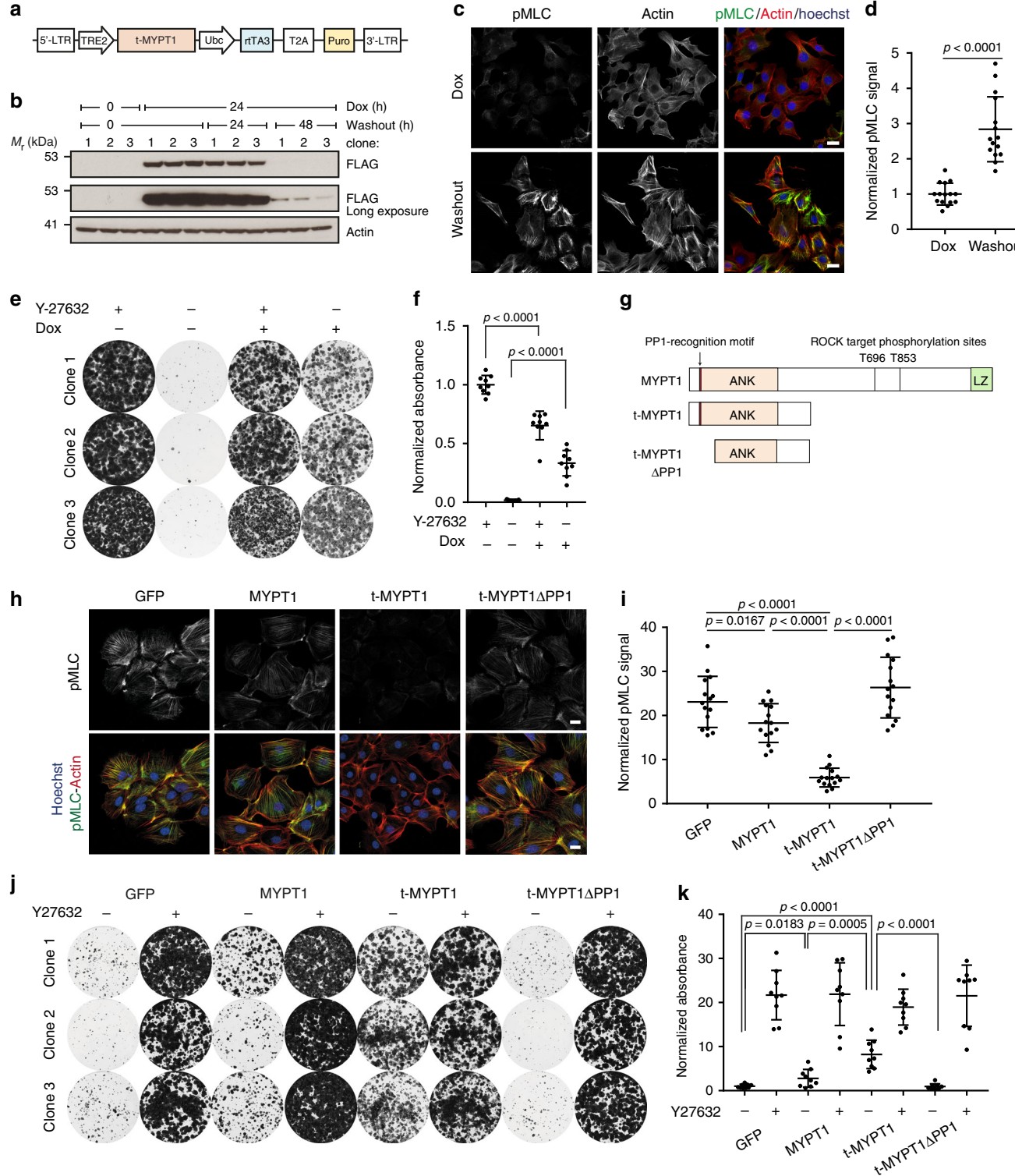

preventing extruded E-cadherin-deficient MMECs from developing into an ILC.

**Relaxation of actomyosin contractility drives ILC formation**. As actomyosin relaxation rescued survival of E-cadherin-deficient MMECs on collagen l, it might also enable the development of ILC, which is typically marked by the massive accumulation of fibrous collagen[8]. We therefore examined whether the relaxation of actomyosin contractility could drive malignant transformation of E-cadherin-deficient MMECs in vivo and promote ILC formation in mice. We performed intraductal injections with lentiviruses encoding MYPT1, t-MYPT1, t-MYPTΔPP1, and empty control in mammary glands of *Wcre;Cdh1^{F/F}* mice and determined tumor burden 20 weeks post-injection (Fig. 7a). As seen before, expression of t-MYPT1 consistently induced formation of classic mouse ILCs characterized by the lack of E-cadherin expression and an Indian file growth pattern in tumor-associated stroma (Fig. 7b, c)[26]. Expression of full-length MYPT1 also induced ILC formation, although tumors formed less frequently

**Fig. 5** MYPT1 drives actomyosin relaxation and cell adhesion. **a** Schematic overview of the doxycycline (dox) inducible t-MYPT1 (MYPT1[1–413]) construct. **b** Western blot analysis of Wcre;Cdh1[F/F];mTmG MMECs transduced with dox-inducible Flag-tagged t-MYPT1 in the absence, presence, or following washout of 2 mg/mL dox as stained for FLAG and actin (loading control). **c, d** IF staining (**c**) and quantification (**d**) of dox-inducible Flag-tagged t-MYPT1 Wcre;Cdh1[F/F];mTmG MMECs cultured in the presence of 2 μg/mL dox or 50 h post washout stained for phospho myosin light chain (pMLC), actin, and Hoechst. Data are of 15 images per condition. **e, f** Representative images (**e**) and quantification (**f**) of clonogenic assays with Wcre;Cdh1[F/F];mTmG MMECs expressing Dox-inducible Flag-tagged t-MYPT1 cultured with or without 2 mg/mL dox and/or 10 μM Y-27632. Data are of three independent experiments with three clones per experiment. **g** Schematic overview of full-length MYPT1, t-MYPT1, and t-MYPT1ΔPP1 (MYPT1[92–413]) displaying the PP1 recognition motif and the ankyrin repeats (ANK) domain, the leucizine zipper domain and inhibitory phosphorylation sites (T696 and T853). **h, i** IF staining (**h**) and quantification (**i**) of Wcre;Cdh1[F/F];mTmG MMECs transduced with MYPT1-wt, t-MYPT1, t-MYPT1ΔPP1, or GFP 48 h post washout of 10 μM Y-27632 and stained for pMLC, actin, and Hoechst. Data are of 15 images per condition. **j, k** Representative images (**j**) and quantification (**k**) of clonogenic assays with Wcre;Cdh1[F/F];mTmG MMECs transduced with t-MYPT1, t-MYPT1ΔPP1 or GFP cultured in the presence or absence of 10 μM Y-27632 (7 days after seeding the cells). Data are of three independent experiments with three clones per experiment. All data are depicted as mean ± standard deviation. All p values were calculated using an unpaired two tailed t-test. Source data are provided as a Source Data file

and were on average smaller in size. These results are in line with our previous observations that full-length MYPT1 displays reduced actomyosin relaxation activity compared to t-MYPT1 (Fig. 5). Importantly, expression of t-MYPTΔPP1, which is unable to induce actomyosin relaxation, failed to induce ILC formation in Wcre;Cdh1[F/F] mice (Fig. 7b). Altogether, these findings demonstrate that PP1-driven relaxation of actomyosin contractility by t-MYPT1 supports survival of E-cadherin-deficient MMECs in the collagen-rich mammary fibrous stroma, resulting in ILC development.

## Discussion

In this study we uncovered the direct consequences of E-cadherin inactivation in MMECs in vivo. We identified that MMECs upon loss of E-cadherin not only extrude to the lumen as previously reported but also towards the basal lamina[8,10]. In contrast to cells that extrude towards the lumen, E-cadherin-deficient MMECs that extrude to the basal lamina do not undergo apoptosis. While extruded E-cadherin-deficient MMECs persist outside the basal lamina they do not form mammary tumors. However, because these cells do persist for a prolonged period of time they have the opportunity to gain additional mutations that allow tumor initiation. The accumulation and persistence of E-cadherin-deficient MMECs in the fibrous stroma might therefore explain the increased susceptibility of CDH1 mutation carriers to develop ILC[36–38].

Increased actomyosin contractility is generally considered to induce tumor cell invasion and migration, thereby promoting progression of established tumors towards metastatic disease[39–41]. In the present study, we demonstrate that actomyosin relaxation promotes ILC development by enhancing adhesion and survival of E-cadherin-deficient MMECs. However, complete inhibition of actomyosin contractility in E-cadherin-deficient MMECs is not tolerated, indicating that there is a fine balance between too much contractility and too little. Taken together these findings indicate that the role of actomyosin contractility during tumorigenesis is context-dependent and may change depending on the micro-environment, tumor type, and tumor stage.

Extrusion of E-cadherin-deficient MMECs into the mammary fibrous stroma coincides with substantial changes in the extra-cellular matrix. Importantly, different ECM components have differential effects on the survival of E-cadherin-deficient MMECs. Survival of these cells is promoted by the basal lamina components, laminin-332 and collagen IV, but not by collagen I, which is the principle component of the fibrous stroma. Survival of E-cadherin-deficient MMECs on a collagen I matrix is critically dependent on actomyosin relaxation. This dependency can be explained by the differential effects of laminin-332 and collagen I on RhoA activity. It has been demonstrated that adhesion of integrin α3β1 to laminin-332 results in increased RAC1 activity, which indirectly inhibits RhoA via multiple mechanisms

including activation of PAK and p190RhoGAP[42–45]. In contrast, adhesion of integrin α2β1 to collagen I has been shown to increase RhoA signaling[35]. Hence, the presence of fibrillar collagen might prevent spreading and proliferation of the extruded E-cadherin-deficient MMECs.

It is however well established that collagen I promotes invasion and migration of established tumor cells[34,46,47]. Moreover, the presence of large amounts fibrillar collagen is a hallmark of ILC[5,8,48]. It is therefore conceivable that collagen I might promote ILC progression once E-cadherin-deficient MECs have acquired mutations that allow for adhesion and survival on collagen I. These mutations might involve alterations that reduce actomyosin contractility or alterations (such as p53 loss) that enable E-cadherin-deficient MECs to survive high levels of actomyosin contractility[49]. Since fibroblasts, the main producers of collagen, infiltrate E-cadherin-deficient neoplastic lesions and massively accumulate in ILC[6], they likely have conflicting roles depending on the stage of ILC development.

Our results show that relaxation of actomyosin contractility by MYPT1 promotes survival of E-cadherin-deficient MMECs on collagen I and leads to ILC formation. Accordingly, amplification of the MYPT1 homolog MYPT2 or hemizygous deletion of MYH9 are frequently observed in human ILC[26]. The lack of homozygous MYH9 deletions in ILC patients supports the notion E-cadherin-deficient cells do not tolerate complete inhibition of actomyosin contractility. Moreover, mutations in RHOA as well as RhoGEF and RhoGAP genes are common in CDH1-mutated diffuse-type gastric cancer[50–52]. At least several of the RHOA mutations have been identified as dominant-negative and lead to RhoA pathway inhibition[53,54]. Also the observed CLDN18-ARHGAP fusions result in inactivation of ROCK signaling and reduction of acto-myosin contractility[54]. Alternatively, PI3K pathway mutations, which are very common in ILC[36–38], may activate the RhoGT-Pase Rac1 via non-canonical PI3K signaling, resulting in RhoA inhibition and reduced actomyosin contractility[45,55]. Altogether, these observations strongly support a dependency of ILC on actomyosin relaxation and raises the question to what extent the formation of E-cadherin-deficient tumors like diffuse gastric carcinoma are similarly dependent on actomyosin relaxation.

In conclusion, this study provides insights into ILC development by uncovering the direct consequences of E-cadherin inactivation in the mouse mammary gland and reveals acto-myosin contractility as a critical barrier to ILC development. Moreover, the discovery of actomyosin regulation as a druggable oncogenic pathway might provide therapeutic opportunities to treat E-cadherin mutated cancers.

## Methods

**Generation of mice.** Wcre;Cdh1[F/F];mTmG mice were generated by intercrossing Wcre;Cdh1[F/F] mice[13] with mTmG reporter mice[14]. Mice were bred onto an FVB/N

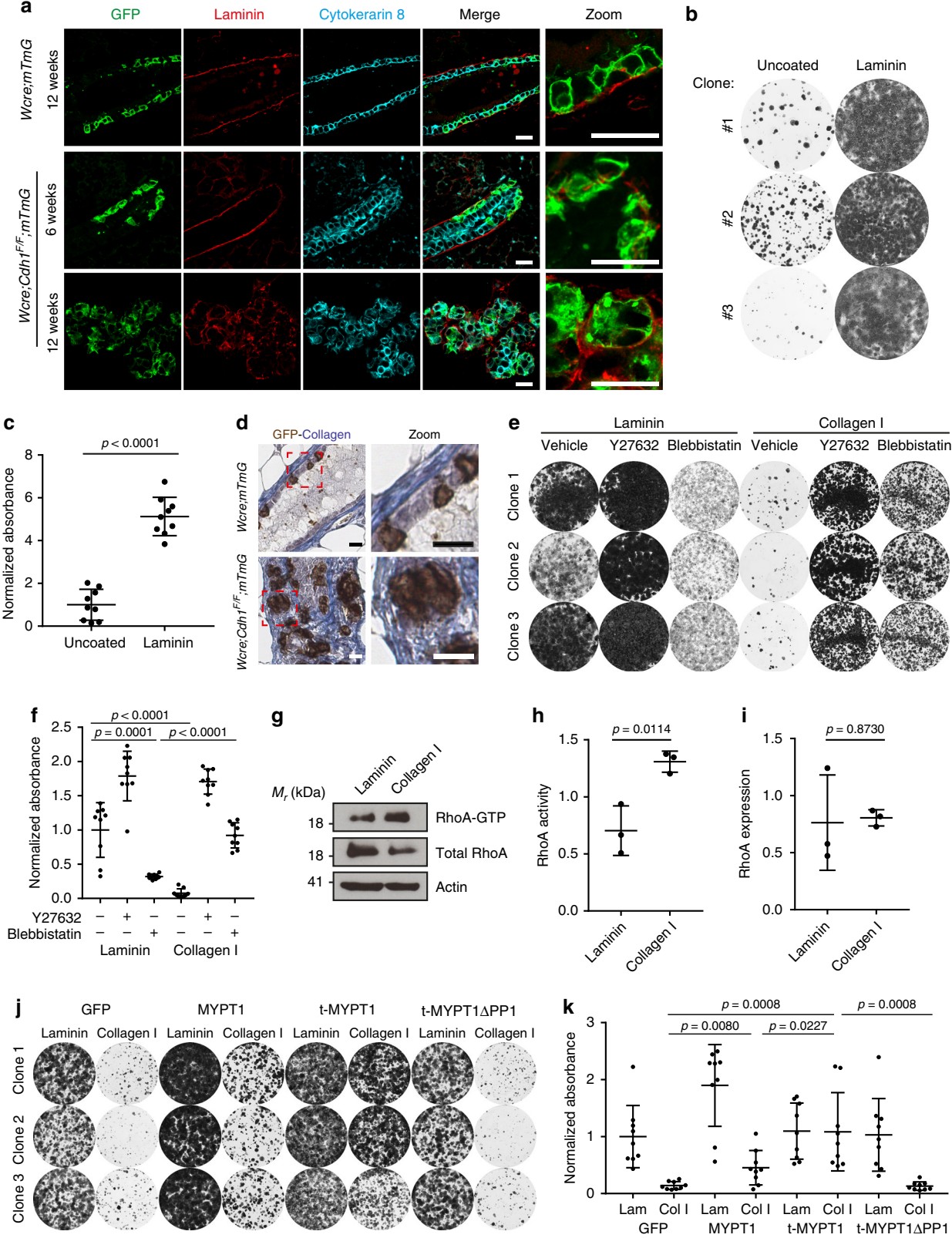

background and genotyped as described[13,14]. All animal experiments were approved by the Animal Ethics Committee of the Netherlands Cancer Institute and performed in accordance with institutional, national, and European guidelines for Animal Care and Use.

**Intravital imaging**. Mice were anesthetized using isoflurane (1.5% isoflurane/ medical air mixture). A small incision was made on top of the ninth and the tenth

mammary gland. The gland was mobilized and placed on a coverslip. The mouse is placed in a facemask with a custom designed imaging box. During imaging the mouse received 100 μL/h Nutrilex special 70/240 (Braun) subcutaneous. Imaging was performed on an inverted Leica SP8 multiphoton microscope with a chameleon Vision-S (Coherent Inc., Santa Clare, CA, www.coherent.com), equipped with four HyD detectors: HyD1 (<455 nm), HyD2 (455–490 nm), HyD3 (500–550 nm), and HyD4 (560–650 nm). Different wavelengths between 800 and 960 nm were used for excitation; collagen (second harmonic generation) was excited with a

**Fig. 6** Collagen I does not support adhesion upon loss of E-cadherin. **a** IF analysis of GFP (green), Laminin (red), and Keratin 8 (cyan) expression in mammary gland sections from 12-week-old *Wcre;mTmG* (top panels) and *Wcre;Cdh1^F/F^;mTmG* mice (middle and bottom panels). Scale bar is 20 μM **b**, **c** Representative images (**b**) and quantification (**c**) of clonogenic assays with *Wcre;Cdh1^F/F^;mTmG* MMEC subclones seeded on laminin-coated wells or uncoated plastic wells. Data are of three independent experiments with three clones per experiment. **d** GFP immunohistochemistry staining combined with a methylene blue staining (collagen). Scale bar is 20 μM. **e**, **f** Representative images (**e**) and quantification (**f**) of clonogenic assays with *Wcre;Cdh1^F/F^; mTmG* MMEC subclones seeded on laminin-coated or collagen I-coated wells in the presence or absence of Y-27632 (10 μM) or blebbistatin (10 μM). Data are of three independent experiments with three clones per experiment. **g–i** Immunoblots (**g**) and quantifications (**h**, **i**) of active RhoA pull-down assays from *Cdh1^Δ/Δ^* MMECs harvested 3 h post plating on laminin- or collagen I-coated plates. Data are of three independent experiments. **j**, **k** Representative images (**j**) and quantification (**k**) of clonogenic assays with *Wcre;Cdh1^F/F^;mTmG* MMECs transduced with t-MYPT1, t-MYPT1ΔPP1, or GFP cultured seeded on laminin-coated or collagen I-coated wells. Data are of three independent experiments with three clones per experiment. All data are depicted as mean ± standard deviation. All *p* values were calculated using an unpaired two tailed *t*-test. Source data are provided as a Source Data file

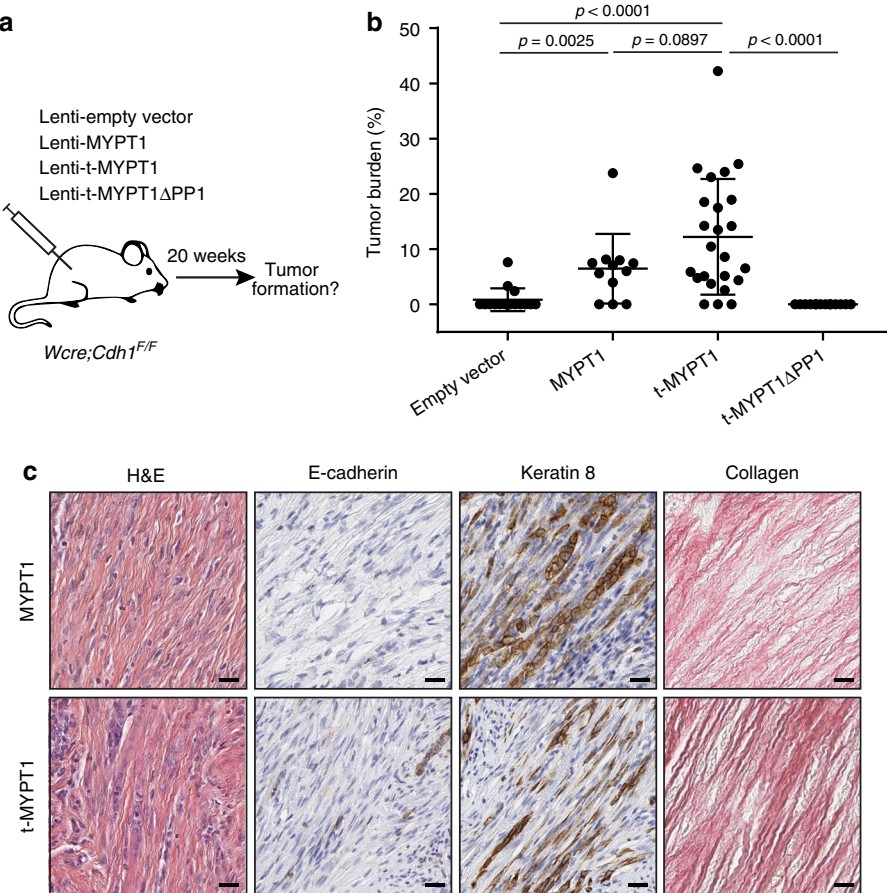

**Fig. 7** Relaxation of actomyosin contractility drives ILC formation. **a** Schematic representation of intraductal injections performed in *WapCre;Cdh1^F/F^* mice with high-titer lentiviruses produced from vectors encoding MYPT1-wt, t-MYPT1, t-MYPT1ΔPP1, or empty vector (EV). **b** Quantified tumor burdens of *Wcre;Cdh1^F/F^* females 20 weeks after intraductal injection of Lenti-EV (*n* = 16 glands), Lenti-MYPT1-wt, (*n* = 12 glands), Lenti-t-MYPT1 (*n* = 24 glands), or Lenti-t-MYPT1ΔPP1 (*n* = 14 glands). Data are depicted as mean ± standard deviation. *p* values were calculated using an unpaired two-tailed *t*-test. **c** Representative images of tumor sections stained with H&E or stained for E-cadherin and CK8 and Collagen (Sirius red). Scale bar is 20 μM. Source data are provided as a Source Data file

wavelength of 800 nm and detected in HyD4. GFP and Tomato were excited with a wavelength of 960 nm and detected in HyD1 and HyD2. Ducts were imaged every 20–30 min using a Z-step size of 5 μm over a minimum period of 5 h. All images were in 12 bit and acquired with a ×25 (HCX IRAPO N.A. 0.95 WD 2.5 mm) water objective. Image analysis was performed manually using Leica application suite X and ImageJ.

**Intraductal injection of lentiviruses**. Intraductal injections were performed as described[56]. Briefly, the mice (2–5 months of age) were anesthetized using keta-mine/sedazine (100 and 10 mg/kg, respectively) and hair was removed in the nipple area with a commercial hair removal cream. Eighteen microliters of high-titer lentivirus mixed with 2 μL 0.2% Evans blue dye in PBS was injected in the fourth mammary glands by using a 34G needle. Mice were handled in a biological safety cabinet under a stereoscope. Lentiviral titers ranging from $2 \times 10^8$ to $2 \times 10^9$ TU/

mL were used. Animals were sacrificed at 20 weeks post-injection. The tumor burden of injected glands was determined as the ratio between the total tumor area and the area of the whole mammary gland, using Fiji (imageJ) software version 1.52n.

**Lentiviral vectors and virus production**. Generation of the SIN.LV.SF-GFP-T2A-puro (GFP, Fig. 4h–l) and SIN.LV.SF-T2A-puro (Empty Vector, Fig. 4o) was described[24]. MYPT1 was isolated with Age1-Sal1 or Age1-Age1 overhangs and a 3′ FLAG tag from cDNA derived from the NMuMG cell line RNA using Phusion Flash High-Fidelity DNA Polymerase (Thermo Scientific). t-MYPT1 was isolated with *Age*1–*Sal*1 or *Bam*H1–*Age*1 overhangs and a 3′ FLAG tag from the *Ppp1r12a*^ex1–9 pBABE puro vector previously described[26] using Phusion Flash High-Fidelity DNA Polymerase (Thermo scientific). The cDNA fragments were then inserted into SIN.LV.SF[57] or SIN.LV.SF-T2A-puro to generate SIN.LV.SF-

MYPT1 SIN.LV.SF-t-MYPT1, SIN.LV.SF-MYPT1-T2A-puro, and SIN.LV.SF-tMYPT1-T2A-puro. MYPT1ΔPP1 was isolated with BamH1–Age1 or BamH1–Sal1 overhangs and a 3′ FLAG tag from SIN.LV.SF tMYPT1 using Phusion Flash High-Fidelity DNA Polymerase. The cDNA fragments were then inserted into SIN.LV.SF or SIN.LV.SF-T2A-puro to generate SIN.LV.SF MYPT1ΔPP1 and SIN.LV.SF-MYPT1ΔPP1-T2A-puro.

The pINDUCER20 (ref. [58]) cassette (with a KRAS[G12D] construct) was isolated with Nhe1 and Sal1 overhangs using Phusion Flash High-Fidelity DNA Polymerase. The cDNA was inserted into a Nhe1 and Sal1 digested SIN.LV.SF vector to generate SIN.LV.SF pINDUCER20 KRAS[G12D]. tMYPT1 was isolated from SIN.LV.SF tMYPT1 with Sal1 and Age1 restriction enzymes. The Age1–Sal1 overhangs were subsequently blunted using Klenow (New England Biolabs). The SIN.LV.SF pINDUCER20 KRAS[G12D] plasmid was digested with EcoRV to remove KRAS[G12D] and inserted with the blunted tMYPT1 cDNA described above to generate SIN.LV.SF pINDUCER20 tMYPT1. All primers are listed in Supplementary Table 1. All vectors were validated by Sanger sequencing. Concentrated lentiviral stocks were produced by transient co-transfection of four plasmids in 293T cells as described[50]. Viral titers were determined using the qPCR lentivirus titration kit from Abm (LV900).

**Cell culture.** Primary MECs were isolated from 10- to 15-week-old females as described[59] and cultured in Dulbecco's modified Eagle's medium (DMEM)-F12 (10565-018; Gibco) with 10% FBS, 1% penicillin–streptomycin, 5 ng/mL epidermal growth factor (EGF) (Sigma), 5 ng/mL insulin (all from Life Technologies) and 5 ng/mL cholera toxin (Gentaur). Y-27632 (10 μM; Abmole, M1817), Blebbistatin (10 μM or as otherwise indicated; Sigma, B0560), or H1152 dihydrochloride (Tocris, 2414) was added to the cell culture medium when indicated. Experiments were performed with primary MEC clones until cell passage 25. Laminin-producing Rac11P cells were cultured in DMEM (Life technologies) with 10% FBS, 1% penicillin–streptomycin.

**Colony formation assays.** Mammary epithelial cells were plated at a density of 5000 cells per plate in six-wells plate in the presence or absence of the indicated drugs (with the indicated concentration). For the inducible t-MYPT1 (Fig. 5c–e) dox was added 24 h before plating to induce expression of t-mypt1 at the time of plating. Seven days after plating the cells were fixated with 4% formaldehyde for 15 min and then stained with 0.5% crystal violet in a 25% methanol solution for at least 30 min. The plates were then washed three times with demi water to remove any remaining dye and dried at room temperature in the dark. Crystal violet stainings were imaged using the Gel count (Oxford Optronix) and its corresponding software. Quantification was performed by dissolving the crystal violet in 10% acetic acid in demineralized water and measuring the absorbance at 590 nm. The absorbance was normalized to the indicated control samples.

**Immunofluorescence.** Formalin-fixed and paraffin-embedded sections were processed as described[6] and incubated overnight at 4 °C with GFP (1:200;Abcam #13970), Cytokeratin-8 (1:200; DSHB Troma-1), Cytokeratin-14 (1:200; Covance PRB-155P), pMLC (1:100; Cell Signaling #3671), E-cadherin (1:200; E-Bioscience #610181), laminin (1:100; Abcam #11575), primary antibodies. Secondary antibodies anti-Rat-Alexa Fluor 647 (1:1000; Invitrogen #A21247), anti-Rabbit-Alexa Fluor 568 (1:1000, Invitrogen #A11011), anti-Rat-Alexa Fluor 568 (1:1000; Invitrogen #11077), anti-mouse-Alexa Fluor 488 (1:1000; Molecular Probes #A21141), anti-Mouse-Alexa Fluor 405 (1:100; Invitrogen #A31553) were incubated overnight at 4 °C. Sections were subsequently stained with Hoechst (1:1000; Thermo Scientific #62249) for 5 min and mounted using Vectashield (Vector Laboratories H-1000). Cultured MMECs were fixed with 4% paraformaldehyde (PFA) and stained with a primary antibody against pMLC (1:100; Cell Signaling #3675), Hoechst (1:1000; Thermos Fischer, 62249), and Alexa Fluor™ 647 Phalloidin (Theremo Fisher, A22287). All images were acquired using a Leica TCS SP5 Confocal and analyzed using LAS AF Version 2.6.3 software. pMLC stainings were quantified using ImageJ and normalized to the GFP surface area.

**Western blot analysis.** Protein lysates were made using one of two lysis buffers: 20 mM Tris-HCl pH 8.0, 150 mM NaCl, 1 mM EDTA, 1 mM EGTA, 1% Triton X-100, 0.5% deoxycholate, 0.1% SDS in milli-Q or 50 mM Tris pH 7.5, 10 mM MgCl$_2$, 0.5 M NaCl, and 2% Igepal (MLC western blots), complemented with protease and phosphatase inhibitors (Roche) and quantified using the BCA protein assay kit (Pierce). Protein lysates were loaded on a 4–12% gel Bis-Tris gradient gel (Invitrogen) and transferred overnight onto nitrocellulose membrane (Bio-Rad) in ×1 transfer buffer (25 mM Tris, 2 M Glycine, 20% methanol in demineralized water). Membranes were blocked in 5% nonfat dry milk or 5% BSA in TBS-T (pH 7.6, 20 mM Tris, 137 mM NaCl, 0.005% Tween-20 in demineralized water) and incubated overnight in 5% nonfat dry milk or 5% BSA in TBS-T with the following primary antibodies: E-cadherin (1:1000; Invitrogen, #13-1900), Cleaved caspase-3 (1:1000; Cell Signaling, #9661), AKT1 (1:1000; Cell Signaling #2938), phospho-AKT(Ser473) (1:1000; Cell Signaling, #4060), FLAG (1:1000; Sigma F7425), MLC (1:1000; Cell Signaling #3672), phospho MLC (Ser20) (1:1000; Abcam #ab2480), Tubulin (1:1000; Sigma, # T9026), and β-actin (1:20,000; Sigma, #A5441). Membranes were washed three times with TBS-T and incubated with one of the following secondary antibodies: anti-Rabbit HRP (1:2000; DAKO, P0260),

Anti-Mouse HRP (1:2000; DAKO, P0448), anti-Rat HRP(1:2000; Invitrogen, 62-9520), or anti-mouse IRDye 680 nm (1:5000; Li-COR 926-32222) in 5% nonfat dry milk or 5% BSA in TBS-T for 1 h at room temperature. Stained membranes were washed three times in TBS-T and then developed using ECL (Pierce 32209 or Bio-Rad 170-5060) on films (Amershan), a Fusion FX7 edge (Vilber) or captured using the Li-Cor Odyssey Infrared Imaging System and analyzed using the Fusion FX7 software or Odyssey Application software version 3.0.16. Uncropped blots can be found in Supplementary Fig. 9.

**Histopathology.** Mouse tissues were formalin-fixed in 10% neutral-buffered formalin for 48 h, embedded in paraffin, sectioned, and stained with hematoxylin and eosin (H&E) or Masson's trichrome. IHC was performed as described[60]. GFP staining was performed by using Rabbit anti-GFP (1:1000; Abcam, ab6556). Secondary antibodies that were used are HRP anti-rabbit Envision (DAKOCytomation, K4011). For the GFP, methyl blue dual staining the GFP staining was performed first the same way as described[8]. The samples were then treated with Bouin's fixative for 1 h at 65°. The samples were counterstained with hematoxiline for 8 min, and rinsed in running tap water for 10 min. Samples were washed in demi water twice for 5 min and exposed to phosphomolybdic acid for 5 min. Phosphomolybdic acid solution was allowed to run of the samples for 5 min and methyl blue solution was added for 5 min. Samples were then washed in demi water for 5 min and washed in 1% acetic acid for 2 min. Samples were then mounted the same as described[9]. All slides were digitally processed using the Aperio ScanScope (Aperio, Vista, CA, USA) and captured using ImageScope software version 12.0.0 (Aperio).

**Flow cytometry.** MEC clones were collected in 1% BSA in 1× PBS and stained with antibodies against: α1 Integrin (1:100; BD Biosciences, 555001), α2 Integrin (1:100; BD Biosciences, 553858), β1 integrin-APC (1:50; Thermo Fischer, 17-0291-82), Cy5-Goat anti Hamster (1:500 Jackson Laboratories), and DAPI (1:20) to exclude dead cells. Samples were analyzed using a BD LSRFortessa Cell Analyzer with BD FACSDiva Software and analyzed using FlowJo software.

**RhoA activation assay.** Fifteen Million cells were plated on regular plastic, collagen-coated or laminin-coated plates and incubated for 3 h at 37°. Cells were washed with PBS and lysed with the lysis buffer included in the RhoA activity assay kit (Cytoskeleton, BK036). Lysates were incubated with 30 μg RhoThekin beads for 1 h at 4° while rotating. Samples were washed once with wash buffer and 20 μL of 2× sample buffer (Invitrogen, NP0007 and NP0009) was added. Samples were incubated at 95° for 5 min, run on NuPAGE™ 4–12% Bis-Tris Protein Gels (Thermo Fisher, NP0321), and transferred overnight on 0.2 μM nitrocellulose membranes (Bio-rad, 1620112) in transfer buffer (25 mM Tris, 2 M glycine, 20% methanol in demineralized water). Membranes were blocked in 5% nonfat dry milk in TBS-T (pH 7.6, 20 mM Tris, 137 mM NaCl, 0.005% Tween-20 in demineralized water). Mouse anti RhoA antibody (1:500; included in kit) was incubated overnight in 0.5% nonfat dry milk in TBS-T. Membranes were washed three times in TBS-T and incubated with the secondary antibody rabbit anti-mouse HRP (1:2000; Dako, P0260) in 5% nonfat dry milk in TBS-T. Stained membranes were washed three times for 10–20 min in TBS-T and subsequently developed using ECL (Pierce, 32209). The signal intensities were measured by densitometry in ImageJ and normalized to actin loading controls.

**Laminin and collagen coating.** The laminin coating was generated by letting RAC-11P[61] cells grown to complete confluence. The plates were washed with 1× PBS and incubated overnight with 20 mM EDTA in 1× PBS overnight. The RAC-11P cells were then removed as sheets by pipetting and washing with 1× PBS. The coated plates were then stored at 4° in 1× PBS until used. Collagen I coating was performed using 100 μG/mL Rat tail Collagen I (Corning, 354249) in 20 mM acetic acid for 2 h at 37° followed by washing with 1× PBS to neutralize the acid. The coated plates were used immediately or stored at 4° until use (no longer than 24 h). Collagen IV coating was performed using 100 μG/mL collagen IV (Santa Cruz, sc-29010) in 0.05 N HCl for 1 h at room temperature followed by washing with 1× PBS to neutralize the acid. The coated plates were used immediately or stored at 4° until use.

**Statistical analyses.** Graphpad Prism v7.03 was used to generate all graphs and perform the statistical analyses. Data are represented as mean and standard deviations. All p values were calculated using an unpaired two tailed t-test.

**Reporting summary.** Further information on research design is available in the Nature Research Reporting Summary linked to this article.

## Data availability

Source data for Figs. 1g,h, 2c, d, f, 3d, j, m, 4b, d, 5d, f, I, k, 6c, e, h, i, k, 7b and Supplementary Figs. 3a-c, 4b and 6b can be found in the source data file. All other data supporting the findings of this study can be obtained from the corresponding authors upon reasonable request.

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

## Acknowledgements

We are grateful to Julia Houthuijzen, Chiara Brambilasca, Sam Cornelissen, Stefano Annunziato, and Marjolijn Mertz for providing technical suggestions and/or help with the experiments. We thank the NKI animal facility, the animal pathology facility, and the digital microscopy facility for their expert technical support. Financial support was provided by the Dutch Cancer society (KWF: project 2015-7589), the Netherlands Organization for Scientific Research (NWO: Cancer Genomics Netherlands (CGCNL), VENI 016156012 (to M.N.) and VICI 91814643 (to J.J)), the European Research Council (ERC Synergy project CombatCancer (to J.J.), ERC consolidator project Cancer-Recurrence 648804 (J.V.R.)) and the Doctor Josef Steiner Cancer Foundation (to J.V.R.). This work is part of the Oncode Institute which is partly financed by the Dutch Cancer Society.

## Author contributions

K.S., M.N., A.S., J.V.R., and J.J. designed experiments. K.S., M.N., D.S., V.R, A.P.D., and E.B. performed experiments. M.N. and J.J. supervised the study. K.S., M.N., and J.J. wrote the manuscript.

## Additional information

**Competing interests:** The authors declare no competing interests.

