## [Peer Review File · Nature Communications]

Reviewers' comments:

Reviewer #1, Expertise: Actomyosin, cancer (Remarks to the Author):

In this work Schipper et al investigate the suppressive role of actomyosin in tumour formation upon loss of E-cadherin in invasive lobular carcinoma. Data is presented in a very biased manner and fails to convince. Much deeper understanding of this process is needed for publication.

Major concerns

1. Authors have a bias on the literature they decide to cite. ROCK and Myosin signalling have been described to have a positive role in tumorigenesis- and a very prominent role in metastatic disease. As a few examples I recommend reading very respected recent work in several tumour types in which ROCK plays a pro-tumorigenic role- such as: Rath, *Cancer Res*, 2018; Rath *Embo mol medicine*, 2018; Vennin *Sci Transl Med*, 2017; Dyberg *PNAS* 2017; Kuemper *e-LIFE*, 2016; Sadok, *Cancer Res*, 2015; Smit *Mol Sys Biol*, 2014; Kumar *Cell* 2012; Samuel, *Cancer Cell*, 2011 and the list goes on....Importantly, this list includes a nice paper from the Derksen lab (and the senior author of this manuscript as a collaboration), in which a clear pro-tumorigenic and pro-metastatic role of ROCK is described in the very same cancer type of the current study -lobular carcinoma. I quote from Schackmann et al, *Journal Clinical Investigation*, 2011 "Primary human ILC samples expressed hallmarks of active Rock signaling, and Rock controlled the anoikis resistance of human ILC cells. Thus, we have linked loss of E-cadherin - an initiating event in ILC development - to Rho/Rock-mediated control of anchorage-independent survival. Because activation of Rho and Rock are strongly linked to cancer progression and are susceptible to pharmacological inhibition, these insights may have clinical implications for the development of tailor-made intervention strategies to better treat invasive and metastatic lobular breast cancer". Why authors do not cite their own work in this context is surprising.

2. Figure 1: figure 1 d only shows a picture, please provide quantification of % extruded cells, same in Supp 1c provide some image analysis quantification, throughout the paper this is a problem- lack of quantitative image analysis. Authors state: the clusters of extruded cells did not seem to increase in size over time: how long is this period of time? authors should provide time scale and proper size measurements.

3. Figure 2: (b,c) membrane blebbing is shown for extruded cells, but nothing else. These cells really look amoeboid and contractile highly migratory cells. Amoeboid behaviour can be quantified from the movies (again all data here is mainly qualitative and not quantitative) figure 2d and supp 2- same applies here, quantification of pMLC is required- and a better description of the structures of the mammary epithelia which looks much more disorganised upon E-cadherin loss. In fact, if loss of E-cadherin induces amoeboid migration which could lead to very early metastatic dissemination (more in accordance with Derksen 2011 work). Have authors considered this possibility?

4. From them on, most of the data presented is based on adding one ROCK inhibitor (Y27632) on the culture media before cells have attached and observing that cells attach better on plastic if there is Y27632 in that media. Y27632 is a ROCK inhibitor of low specificity and low potency. There are much better inhibitors to choose from (Feng et al, *J Med Chem*, 2016), such as GSK269962 or H1152 to target Myosin II activity completely.

5. Furthermore, to discard drug off target effects of Y27632, genetic depletion of either ROCK1/2 and Myosin II (MLC2) should be provided. Authors should also let the cells attach first and then add the ROCK inhibitors or Myosin II inhibitor. That would resolve the question if Myosin plays a role only in attachment to the plastic or in proliferation -after cells have attached (even if it is a very artificial substrate: plastic). Also, to achieve total inhibition of Myosin II higher concentration of Blebbistatin are needed. On the same lines, how often are these inhibitors added (methods are only poorly described)

6. Figure 3: showing one Rho pulldown is insufficient, there is no information of how many times the experiments were done? Quantification? Statistical analysis? From the one blot they show it looks like it is the expression levels that are regulated (rather than activity)

7. Taking into account previous findings of the Jonkers lab that some ILC harbour mutations in MYPT and importantly in MYH9, this reviewer interprets this data differently. 1st MYH9 mutations

are always heterozygous, that is cells are selected to retain some myosin II activity. This could indicate that cells need the right amount of contractility to survive when they lose E-Cadherin. Total loss of contractility is probably deleterious (like other very many studies have shown). But possibly too much contractility is also incompatible with strong levels of adhesion. Therefore, an intermediate level of Myosin II (somehow like a rheostat) is achieved (that is what also could happen with Y27632 -which is not very potent, or with 10 micromolar blebbistatin). In fact, mutations in MYPT could be indicative of that. There are more than half a dozen of kinases that phosphorylate Myosin II (ROCK, MRCK, MLCK, ZIPK, Citron kinase, CaMK...) why not inactivate those? Instead cells keep the kinases active but mutate the phosphatase so the levels can be tuned to the right levels. Authors should consider this alternative hypothesis- that would unify better current findings in cancer, it is not total loss of Myosin II but the right levels of Myosin II.

8. Figure 4a. It is strange that pictures show "0" signal for Myosin II activity but then quantifications are normalised to value "1". This is not possible- there has to be some level of signal to be able to give "1" as a value.

9. All the experiments with truncated MYPT are lacking the control of full length MYPT – this controls are crucial since MYPT is negatively regulated by ROCK (in residue Thr 696) and this residue seems to be lacking from the constructs they are using (even if not much information is provided). This full length is important to understand the real mechanism since it is ROCK regulated. In those lines, what happens to MLC activity in 4k,l? it is odd that there is no synergy between MYPT truncation and ROCK inhibitor: in fact it seems like MYPT+Y27632 has less colonies than Y27632? Truncated MYPT kills cells in the context of ROCK inhibition? (Figure 4f and 4i) this does not support their hypothesis.

10. In vivo work is insufficient: authors should use ROCK inhibitors/depletion and full length MYPT to support their data

11. Figure 5. What is the effect of ROCK inhibition in these more relevant matrices (collagen and laminin)? Why base the whole paper on plastic as a substrate to recapitulate a survival signal coming from either collagen or laminin? odd experimental choice. Main collagen receptor is beta 1 integrin – what are the levels? There is evidence in the literature that Myosin II (via YAP) provides a survival signal in these substrates.

12. Figure 5d, are these cells alive? If collagen allows those cells to expand in that niche and later invade -maybe these are the dangerous cells?

13. Figure 5g. Pull down is of low quality, number of times that experiments were performed are not included and there is no quantification and no statistical analysis.

Minor comments

1. Discussion: Page 8: "collagen I might promote ILC progression once E-Cadherin deficient MMECs have acquired (epi) genetic alterations to reduce actomyosin contractility" This is highly unlikely since they have already described with Dreksen lab "Primary human ILC samples expressed hallmarks of active Rock signalli

Reviewer #2, Expertise: breast cancer (Remarks to the Author):

This manuscript is a logical follow up to the laboratory's previous publication (Kas et al., 2017 Nat Genetics) where they using an in vivo sleeping beauty screen to identify secondary mutations that drive invasive lobular carcinoma (ILC) formation in mice with targeted deletions in E-cadherin. One of the identified hits truncating mutations in the Ppp1r12a gene (encoding myosin phosphatase target subunit 1, MYPT1). Here the authors, make two key observations. The first, novel observation (derived for a variety of careful experimental in vitro and in vivo approaches), is that E-cadherin-defective cells are extruded and localise to the basal laminal border where they remain viable and that this is mediated, at least in part, by the interaction with the basal lamina component laminin-332. The second, unites this observation with their previous findings to demonstrate that impairment of actomyosin contractility (as demonstrated by subsequent expression of truncated MYPT1) drives the development of ILC in mice.

Taking these data together the authors propose that the inability of E-cadherin-defective cells to move out of their basal niche and invade into the surrounding tissue is due to reduced viability (due to increased actomyosin contractility) when interacting with collagen and/or loss of contact with laminin. In support of this they demonstrate that when plated on collagen the cells show increased RhoA activation compared to cells plated onto laminin and conclude that secondary mutation (or epigenetic changes) that limit actomyosin contractility are required to overcome this limitation.

Overall this is a very high-quality manuscript with convincing data (and a substantial amount of data) and providing important new insight in the development of lobular breast cancers (15% of all breast cancers). That said, there is one piece of the evidence missing that the authors should either address experimentally or within the text of the manuscript. Personally, I think there is sufficient new (and high quality) data presented here to warrant publication and as a consequence I would be satisfied if this was just addressed in the text.

Major comments

The authors have demonstrated that a collagen substratum does not support the adhesion and survival of their E-cadherin-defective mouse mammary epithelial cells (Fig. 5e,f) but they have not demonstrated as stated on Line 166 that collagen hampers their survival as there is no difference between uncoated plates and collagen coated plates. Moreover, they provide no data to demonstrate cells expressing the truncated-MYPT1 show enhanced survival on collagen plates. It may be that the authors have performed this experiment and not seen a rescue - if that is the case, then the conclusion of the manuscript with regards to collagen should be toned down.

Minor comments

Line 66 - Should be Fig 2b,d

Line 74 & 76 - Should be Fig. 2e

Figure 4n - lacks a scale bar and needs state in the legend what stain was used to detect collagen.

Reviewer #3, Expertise: matrix, cancer (Remarks to the Author):

In the submitted paper the authors have investigated the outcome of E-cadherin deletion in luminal MMECs and investigated the distinction between E-cadherin loss, which results in impaired survival, and the ILC (a specific form of breast cancer characterized by E-cadherin loss). Using a mouse model of conditional deletion of E-cadherin specifically in the K8-positive luminal cells, they find that E-cadherin null cells are extruded basally and can breach the basal lamina. However, a second step, relaxation of the acto-myosin contractility is needed for the cells to survive in collagen and gain ability to disseminate. Hence they define that cell intrinsic contractility acts as a barrier for ILC development. These data are interesting as they provide important some new insight into this poorly understood form of breast cancer. However, in its current form this is a rather descriptive study with limited mechanistic insight. In addition, many experiments lack proper quantification and the authors should consider state-of-the art imaging like FRET-probes for RhoA activity in vitro and in vivo to strengthen their conclusions.

Fig 1. It would be informative if the stroma (1D) and basal lamina (1E) could be visualized as well in addition to the cells.

Fig 2E. The pMLC levels would need to be quantified from several sections/animals and numerous cell clusters to provide some quantitative insight into the claimed elevation of pMLC in the GFP-positive E-cadherin-deficient cells.

Fig 3D,E (3I, J) Can the authors show that pMLC is elevated specifically in the deficient blebbing MMECs? The data in 3k is hard to interpret as the right lane has higher loading and both total

Rhoa and RhoGTP as well as MLC and pMLC are elevated. These data need to be shown from several experiments and quantified to validate the claim of elevated RhoA activity (the same point is relevant also for 5G). What about other Rho proteins?

Fig. 5D. Please include a IF staining of collagen and high resolution images

Does ROCK inhibition support proliferation of the cells on Type I collagen? What is the response of the cells to basal lamina component Col IV. What is the mechanism for the distinct effects of contractility on ColI and Laminin?

We would like to thank the reviewers for their constructive feedback. We believe that thanks to their comments the manuscript has significantly improved. The manuscript was converted from a letter format into a full article and the number of figures has increased from 5 to 7. To minimize confusion, we have included the original figure references in our point-by-point responses.

Reviewer #1, Expertise: Actomyosin, cancer(Remarks to the Author):

In this work Schipper et al investigate the suppressive role of actomyosin in tumour formation upon loss of E-cadherin in invasive lobular carcinoma. Data is presented in a very biased manner and fails to convince. Much deeper understanding of this process is needed for publication.

Major concerns

1. Authors have a bias on the literature they decide to cite. ROCK and Myosin signalling have been described to have a positive role in tumorigenesis- and a very prominent role in metastatic disease. As a few examples I recommend reading very respected recent work in several tumour types in which ROCK plays a pro-tumorigenic role- such as: Rath, Cancer Res, 2018; Rath Embo mol medicine, 2018; Vennin Sci Transl Med, 2017; Dyberg PNAS 2017; Kuemper e-LIFE, 2016; Sadok, Cancer Res, 2015; Smit Mol Sys Biol, 2014; Kumar Cell 2012; Samuel , Cancer Cell, 2011 and the list goes on....Importantly, this list includes a nice paper from the Derksen lab (and the senior author of this manuscript as a collaboration), in which a clear pro-tumorigenic and pro-metastatic role of ROCK is described in the very same cancer type of the current study -lobular carcinoma. I quote from Schackmannet al, Journal Clinical Investigation, 2011 "Primary human ILC samples expressed hallmarks of active Rock signaling, and Rock controlled the anoikis resistance of human ILC cells. Thus, we have linked loss of E-cadherin - an initiating event in ILC development - to Rho/Rock-mediated control of anchorage-independent survival. Because activation of Rho and Rock are strongly linked to cancer progression and are susceptible to pharmacological inhibition, these insights may have clinical implications for the development of tailor-made intervention strategies to better treat invasive and metastatic lobular breast cancer". Why authors do not cite their own work in this context is surprising.

Reply: We agree with the reviewer that ROCK activity can have a pro-tumorigenic role in various cancer types including ILC. The main difference between the above-mentioned studies and the current study is the stage of tumorigenesis. The above-mentioned studies were all performed with cell lines derived from established tumors whereas our study is done in primary non-transformed murine mammary epithelial cells (MMECs). Hence, our study focuses on the effect of actomyosin contractility on tumor initiation rather than on tumor progression. We therefore believe that our study does not contradict the above mentioned studies but rather identifies an opposing role for actomyosin contractility during ILC initiation. We do agree with the reviewer that this dual role was not clearly explained in the original manuscript. We have therefore added a section in the discussion of the revised manuscript to emphasize the opposing role of actomyosin contractility in ILC (lines 286-294 of the revised manuscript).

2. Figure 1: figure 1 d only shows a picture, please provide quantification of % extruded cells, same in Supp 1c provide some image analysis quantification, throughout the paper this is a problem- lack of quantitative image analysis. Authors state: the clusters of extruded cells did not seem to increase in size over time: how long is this period of time? authors should provide time scale and proper size measurements.

Reply: We have quantified the amount of extruded cells in 3-month-old *Wcre;Cdh1^{F/F};mTmG* mice compared to age-matched control mice (Figure 1f). Furthermore, we have quantified the size of GFP positive clusters over a time period of 3 months to 1 year (Figure 1g).

During revision of our manuscript we concluded that assessment of ZO1 expression (shown in Supplementary Figure 1C of the original manuscript) is not informative since our study does not address the potential role of tight junctions in the E-cadherin-deficient clusters. We therefore decided to remove supplementary figure 1c and the accompanying statement in the text.

3. Figure 2: (b,c) membrane blebbing is shown for extruded cells, but nothing else. These cells really look amoeboid and contractile highly migratory cells. Amoeboid behaviour can be quantified from the movies (again all data here is mainly qualitative and not quantitative) figure 2d and supp 2- same applies here, quantification of pMLC is required- and a better description of the structures of the mammary epithelia which looks much more disorganised upon E-cadherin loss. In fact, if loss of E-cadherin induces amoeboid migration which could lead to very early metastatic dissemination (more in accordance with Derksen 2011 work). Have authors considered this possibility?

Reply: We agree that our description of the behavior of E-cadherin-deficient MMECs *in situ* was insufficient. We therefore included a more detailed description of the behavior of these cells in the results section (marked text line 98-102). In line with the reviewer, we initially hypothesized that loss of E-cadherin might induce early cell dissemination. We therefore specifically looked for disseminating cells in *Wcre;Cdh1^{F/F};mTmG* mammary glands by intravital imaging. Moreover by IHC staining of GFP we also looked for disseminated cells in the surrounding stroma and even other tissues of *Wcre;Cdh1^{F/F};mTmG* mice, but never encountered GFP-marked E-cadherin deficient disseminated cells. Intravital imaging of E-cadherin-deficient cells in *Wcre;Cdh1^{F/F};mTmG* mammary glands revealed that E-cadherin-deficient MMECs are motile but do not migrate or display amoeboid behavior. Instead, these cells seem to tumble around each other. Altogether our observations indicate that E-cadherin-deficient mammary epithelial cells do not disseminate but persist in the laminin-rich fibrous stroma for at least one year (Figure 1e), It is conceivable that these cells may over time acquire additional mutations that promote tumor formation and progression.

To address the second point of the reviewer we have quantified the amount of GFP-positive pMLC^{high} cells in figure 2f (please see the response to reviewer 3 comment 2 for more details).

4. From them on, most of the data presented is based on adding one ROCK inhibitor (Y27632) on the culture media before cells have attached and observing that cells attach better on plastic if there is Y27632 in that media. Y27632 is a ROCK inhibitor of low specificity and low potency. There are much better inhibitors to choose from (Feng et al, J Med Chem, 2016), such as GSK269962 or H1152 to target Myosin II activity completely.

Reply: We have performed a colony formation assay with a range of concentrations of H1152 demonstrating that this inhibitor, like Y27632, rescues cell adhesion of E-cadherin-deficient MMECS, albeit at a much lower concentration (figure 4b). This is in line with the increased specificity for ROCK 1/2 of H1152 compared to Y27632.

5. Furthermore, to discard drug off target effects of Y27632, genetic depletion of either ROCK1/2 and Myosin II (MLC2) should be provided. Authors should also let the cells attach first and then add the ROCK inhibitors or Myosin II inhibitor. That would resolve the question if Myosin plays a role only in attachment to the plastic or in proliferation -after cells have attached (even if it is a very artificial substrate: plastic). Also, to achieve total inhibition of Myosin II higher concentration of Blebbistatin are needed. On the same lines, how often are these inhibitors added (methods are only poorly described)

Reply: Complete inhibition of ROCK1/2, MYH9 (as observed previously (Kas et al. 2017¹) and MLC is not tolerated and genetic inactivation would therefore not result in stable cell lines to test. Y27632 or blebbistatin cannot be added after the cells attach since they die before this point. We included a detailed description of the colony formation experiments in the methods section of the revised manuscript. We performed additional colony formation assays with a concentration range of Blebbistatin. The results shown in figure 4a,b of the revised manuscript indicate that higher levels of blebbistatin are not tolerated and result in reduced survival.

6. Figure 3: showing one Rho pulldown is insufficient, there is no information of how many times the experiments where done? Quantification? Statistical analysis? From the one blot they show it looks like it is the expression levels that are regulated (rather than activity)

Reply: We have conducted the pulldown experiments three times and have quantified the results which are show in figure 3m. We agree with the reviewer that the effect of E-cadherin loss on RhoA activity seems to be mainly due to increased expression levels of RhoA.

7. Taking into account previous findings of the Jonkers lab that some ILC harbor mutations in MYPT and importantly in MYH9, this reviewer interprets this data differently. 1st MYH9 mutations are always heterozygous, that is cells are selected to retain some myosin II activity. This could indicate that cells need the right amount of contractility to survive when they lose E-Cadherin. Total loss of contractility is

probably deleterious (like other very many studies have shown). But possibly too much contractility is also incompatible with strong levels of adhesion. Therefore, an intermediate level of Myosin II (somehow like a rheostat) is achieved (that is what also could happen with Y27632 -which is not very potent, or with 10 micromolar blebbistatin). In fact, mutations in MYPT could be indicative of that. There are more than half a dozen of kinases that phosphorylate Myosin II (ROCK, MRCK, MLCK, ZIPK, Citron kinase, CaMK...) why not inactivate those? Instead cells keep the kinases active but mutate the phosphatase so the levels can be tuned to the right levels. Authors should consider this alternative hypothesis- that would unify better current findings in cancer, it is not total loss of Myosin II but the right levels of Myosin II.

Reply: We completely agree with the reviewer that survival of E-cadherin-deficient cells is fostered by dosage reduction rather than complete abolishment of actomyosin contraction. We agree that the initial manuscript did not adequately address this important issue. We have therefore performed additional experiments with a concentration range of blebbistatin and the selective ROCK inhibitor H1152. The results of these experiments are shown in figure 4a-d of the revised manuscript and demonstrate that higher concentrations of blebbistatin and H1152 hamper cell survival compared to the lower doses. To emphasize the need for balanced actomyosin contractility we have added an additional section in the results (lines 166 to 182 in the revised manuscript) and included an additional section in the discussion (lines 286-294 in the revised manuscript).

8. Figure 4a. It is strange that pictures show "0" signal for Myosin II activity but then quantifications are normalised to value "1". This is not possible- there has to be some level of signal to be able to give "1" as a value.

Reply: We agree that it was difficult to see the signal in the provided example. This was due to the switch from CMYK to RGB and a reduced resolution of the original PDF file. We have increased the upper threshold on the pMLC channel in both the washout and continuous dox which should make it clearer that there is still pMLC signal in the continuous DOX treated sample.

9. All the experiments with truncated MYPT are lacking the control of full length MYPT – this controls are crucial since MYPT is negatively regulated by ROCK (in residue Thr 696) and this residue seems to be lacking from the constructs they are using (even if not much information is provided). This full length is important to understand the real mechanism since it is ROCK regulated. In those lines, what happens to MLC activity in 4k,l? it is odd that there is no synergy between MYPT truncation and ROCK inhibitor: in fact it seems like MYPT+Y27632 has less colonies than Y27632? Truncated MYPT kills cells in the context of ROCK inhibition? (Figure 4f and 4i) this does not support their hypothesis.

Reply: The reviewer is correct in remarking that the ROCK phosphorylation sites are not present in truncated MYPT1 and therefore it cannot be inhibited by ROCK. We have added the relevant phosphorylation sites in the schematic overview (Figure 5g). We also agree with the reviewer that the addition of full length MYPT1 provides important information. We have therefore included full-length

MYPT1 as a control in all experiments with truncated MYPT1 (figure 5h-k). This shows as expected that full length MYPT1 gives an intermediate phenotype, indicating that full-length MYPT1 is partially inhibited by Rho Kinase. We have also measured the effect of the various MYPT1 variants on MLC activity in figure 4K,I (now Fig. 5h to 5i). Given the need for a balance in actomyosin contractility (as the reviewer suggests in comment 7) it makes sense that a combination of truncated MYPT1 and Y-27632 reduces proliferation.

10. In vivo work is insufficient: authors should use ROCK inhibitors/depletion and full length MYPT to support their data

Reply: We have included full-length MYPT1 in our in vivo study and added the results to figure 7 of the revised manuscript. These new data show that (in line with our in vitro experiments) full-length MYPT1 gives an intermediate phenotype, which is likely due to the fact that full-length MYPT1 can still be inhibited by ROCK. Although it will be interesting to investigate the effect of genetic or pharmacological inhibition of Rho kinase on *de novo* tumor development in *Wcre;Cdh1^{F/F}* female mice, these experiments are both challenging and time-consuming, and therefore beyond the scope of this paper.

11. Figure 5. What is the effect of ROCK inhibition in these more relevant matrices (collagen and laminin)? Why base the whole paper on plastic as a substrate to recapitulate a survival signal coming from either collagen or laminin? odd experimental choice. Main collagen receptor is beta 1 integrin – what are the levels? There is evidence in the literature that Myosin II (via YAP) provides a survival signal in these substrates.

Reply: We agree that it would be informative to know the effect of ROCK/myosin inhibition on Collagen and Laminin matrices. We therefore repeated the experiment shown in figure 5e,f of the original manuscript with addition of Y27632 and blebbistatin. The results of these experiments, which can be found in figure 6d,e of the revised manuscript, indicate that both Rho Kinase and myosin inhibition can rescue cell adhesion on a Collagen matrix.

We determined the expression of beta 1 integrin next to alpha 1 and alpha 2 in our E-cadherin-deficient and WT MMECs and found both beta 1 and alpha 2 are highly expressed while alpha 1 integrin is expressed in neither E-cadherin-deficient nor WT MMECs (Supplementary Fig 7).

12. Figure 5d, are these cells alive? If collagen allows those cells to expand in that niche and later invade -maybe these are the dangerous cells?

Reply: These cells are alive but since the cluster size does not increase over time (figure 1h) and since collagen I does not mediate adhesion and survival, we hypothesize that the collagen surrounding the

clusters is not allowing them to expand. We do agree that these cells are nevertheless dangerous as they may acquire additional mutations that promote their progression into ILC.

13. Figure 5g. Pull down is of low quality, number of times that experiments were performed are not included and there is no quantification and no statistical analysis.

Reply: We have conducted the pulldown three times and have quantified the results, which are shown in figure 6i. We observed that there is a significant increase in active RhoA in E-cadherin-deficient MMECs on a collagen I matrix compared to a laminin matrix.

Minor comments

1. Discussion: Page 8: "collagen I might promote ILC progression once E-Cadherin deficient MMECs have acquired (epi) genetic alterations to reduce actomyosin contractility" This is highly unlikely since they have already described with Dreksen lab "Primary human ILC samples expressed hallmarks of active Rock signaling

Reply: We agree with the reviewer that reduction of actomyosin contractility is not a uniform feature of all ILCs. We hypothesize that some ILCs may have acquired mutations to reduce actomyosin contractility, whereas others have acquired mutations (such as p53 loss) that enable E-cadherin-deficient MECs to survive high levels of actomyosin contractility. We have therefore altered the statement in the discussion.

Reviewer #2, Expertise: breast cancer (Remarks to the Author):

This manuscript is a logical follow up to the laboratory's previous publication (Kas et al., 2017 Nat Genetics) where they using an in vivo sleeping beauty screen to identify secondary mutations that drive invasive lobular carcinoma (ILC) formation in mice with targeted deletions in E-cadherin. One of the identified hits truncating mutations in the Ppp1r12a gene (encoding myosin phosphatase target subunit 1, MYPT1). Here the authors, make two key observations. The first, novel observation (derived for a variety of careful experimental in vitro and in vivo approaches), is that E-cadherin-defective cells are extruded and localise to the basal laminal border where they remain viable and that this is mediated, at least in part, by the interaction with the basal lamina component laminin-332. The second, unites this observation with their previous findings to demonstrate that impairment of actomyosin contractility (as demonstrated by subsequent expression of truncated MYPT1) drives the development of ILC in mice.

Taking these data together the authors propose that the inability of E-cadherin-defective cells to move out of their basal niche and invade into the surrounding tissue is due to reduced viability (due to

increased actomyosin contractility) when interacting with collagen and/or loss of contact with laminin. In support of this they demonstrate that when plated on collagen the cells show increased RhoA activation compared to cells plated onto laminin and conclude that secondary mutation (or epigenetic changes) that limit actomyosin contractility are required to overcome this limitation.

Overall this is a very high-quality manuscript with convincing data (and a substantial amount of data) and providing important new insight in the development of lobular breast cancers (15% of all breast cancers). That said, there is one piece of the evidence missing that the authors should either address experimentally or within the text of the manuscript. Personally, I think there is sufficient new (and high quality) data presented here to warrant publication and as a consequence I would be satisfied if this was just addressed in the text.

Major comments

The authors have demonstrated that a collagen substratum does not support the adhesion and survival of their E-cadherin-defective mouse mammary epithelial cells (Fig. 5e,f) but they have not demonstrated as stated on Line 166 that collagen hampers their survival as there is no difference between uncoated plates and collagen coated plates. Moreover, they provide no data to demonstrate cells expressing the truncated-MYPT1 show enhanced survival on collagen plates. It may be that the authors have performed this experiment and not seen a rescue - if that is the case, then the conclusion of the manuscript with regards to collagen should be toned down.

Reply: We agree with the reviewer that there is no difference in survival of E-cadherin-deficient MMECs on uncoated and collagen-coated plates. We have therefore altered the statement to: "In contrast to laminin-332, collagen I did not support adhesion and survival of E-cadherin-deficient MMECs despite expression of the collagen-receptor integrin $\alpha 2\beta 1$ (Fig. 6e, f and Supplementary Figure 7)".

We also agree with the reviewer that it is important to test whether E-cadherin-deficient cells expressing the truncated-MYPT1 show enhanced survival on collagen plates. To address this issue we have assessed the effects of truncated-MYPT1, blebbistatin and the Rock inhibitor Y-27632 on growth of E-cadherin-deficient MMECs on a collagen matrix (Figure 6d, e, j, k). Similar to blebbistatin and Y-27632, expression of truncated MYPT1 (and to a lesser extent full-length MYPT1) allows for the survival of E-cadherin-deficient MMECs on a collagen 1 matrix.

Minor comments

Line 66 - Should be Fig 2b,d

Reply: We thank the reviewer for pointing out the incorrect figure references in the text, which has been corrected in the revised manuscript.

Line 74 & 76 - Should be Fig. 2e

Reply: We thank the reviewer for pointing out the incorrect figure references in the text, which has been corrected in the revised manuscript.

Figure 4n - lacks a scale bar and needs state in the legend what stain was used to detect collagen.

Reply: We thank the reviewer for pointing this out. We have added a scale bar and the staining used to detect collagen (Sirius red) in the figure legend.

Reviewer #3, Expertise: matrix, cancer (Remarks to the Author):

In the submitted paper the authors have investigated the outcome of E-cadherin deletion in luminal MMECs and investigated the distinction between E-cadherin loss, which results in impaired survival, and the ILC (a specific form of breast cancer characterized by E-cadherin loss). Using a mouse model of conditional deletion of E-cadherin specifically in the K8-positive luminal cells, they find that E-cadherin null cells are extruded basally and can breach the basal lamina. However, a second step, relaxation of the acto-myosin contractility is needed for the cells to survive in collagen and gain ability to disseminate. Hence they define that cell intrinsic contractility acts as a barrier for ILC development. These data are interesting as they provide important some new insight into this poorly understood form of breast cancer. However, in its current form this is a rather descriptive study with limited mechanistic insight. In addition, many experiments lack proper quantification and the authors should consider state-of-the art imaging like FRET-probes for RhoA activity *in vitro* and *in vivo* to strengthen their conclusions.

We agree with the reviewer that the use of FRET probes would be a very elegant way of demonstrating differential RhoA activity in E-cadherin-deficient MMECs. We have made several attempts to measure RhoA activity in E-cadherin-proficient and -deficient MMECs *in vitro* using a published RhoA FRET-probe² but we were unable to obtain reproducible results. We therefore decided to focus on the RhoA pull-down experiments.

The use of FRET-probes for *in vivo* imaging of RhoA activity would be very time-consuming as it would require (i) import of the mice carrying the RhoA FRET-probe, (ii) crossing of these mice onto our *Wcre;Cdh1^{F/F}* strain and (iii) optimization of the imaging. We therefore believe that this experiment is beyond the scope of the current paper.

Fig 1. It would be informative if the stroma (1D) and basal lamina (1E) could be visualized as well in addition to the cells.

Reply: Unfortunately, in the experiment shown in figure 1d we had no available channels to also visualize stromal components. In figure 6a (figure 5a in the original manuscript), we have visualized the basal

membrane by staining for laminin. Additionally, in figure 6d (figure 5d in the original manuscript) we have performed histochemical staining with methyl blue and IHC with an anti-GFP antibody to visualize collagen fibers and GFP-positive E-cadherin-deficient cells, respectively.

Fig 2E. The pMLC levels would need to be quantified from several sections/animals and numerous cell clusters to provide some quantitative insight into the claimed elevation of pMLC in the GFP-positive E-cadh^{-/-} cells.

Reply: We agree that the manuscript would benefit from a more thorough quantification of the amount of E-cadherin-deficient luminal epithelial cells with increased MLC phosphorylation. Therefore, we have quantified the amount of GFP-positive pMLC^{high} cells in multiple cell clusters from 12 glands and 3 mice per condition.

Fig 3D,E (3I, J) Can the authors show that pMLC is elevated specifically in the deficient blebbing MMECS? The data in 3k is hard to interpret as the right lane has higher loading and both total RhoA and RhoGTP as well as MLC and pMLC are elevated. These data need to be shown from several experiments and quantified to validate the claim of elevated RhoA activity (the same point is relevant also for 5G). What about other Rho proteins?

Reply: To address these valid concerns, we have quantified the RhoA pulldowns shown in figure 3K and figure 6g (figure 5g of the original ms). The results of these quantifications are shown in figures 3m, S3 and 6h,i. The E-cadherin-deficient MMECs have higher expressions of both RhoA as well MLC. The mechanism of how loss of E-cadherin results in increased expression of RhoA and MLC remains to be determined. Nevertheless, our results indicate that the loss of E-cadherin increases the overall contractility which correlates with an increase of total RhoA and pMLC activity.

Fig. 5D. Please include a IF staining of collagen and high resolution images

Reply: A collagen I IF staining is not possible on our formalin fixed tissues and isolating new fresh frozen tissues would be challenging (due to high fat content of mammary glands) and time consuming. We were however able to detect fibrillar collagen 1 using 2nd harmonic generation during the intravital imaging. We have prepared a new figure from images taken during the intravital imaging (Supplementary Fig 6) which clearly shows that the clusters of extruded E-cadherin-deficient MMECs are surrounded by fibrillar collagen.

Does ROCK inhibition support proliferation of the cells on Type I collagen?

Reply: This is an interesting question which was also raised by reviewer 1, please see our response to comment 11 of reviewer 1.

What is the response of the cells to basal lamina component Col IV.

Reply: We agree with the reviewer that this is an interesting question. We have therefore performed colony formation assays with E-cadherin-deficient MMECs on Collagen IV coated plates. Although we observed a survival benefit, this was very modest compared to the survival on laminin 332 coated plates (Figure S5 and 6b, c of the revised manuscript).

What is the mechanism for the distinct effects of contractility on Coll and Laminin?

Reply: We agree with the reviewer that it is important to discuss the possible mechanisms underlying the differential effects of laminin and collagen I on RhoA activity in E-cadherin-deficient MMECs. Previous work by Zhou and Kramer³ has already shown that laminin 332 and collagen 1 have differential effects on RhoA activity in squamous carcinoma cells. This differential effect was found to be due to differential integrin activation by laminin 332 and collagen, respectively. Collagen I adhered to integrin $\alpha 2\beta 1$ leading to activation of RhoA, while laminin 332 adhered to integrin $\alpha 3\beta 1$ resulting in the activation of CDC42 and Rac1. It is well-established that activation of Rac1 indirectly inhibits RhoA⁴⁻⁷. We have added a section in the discussion to address this specifically (Lines 300-306 of the revised manuscript).

References

1. Kas, S. M. *et al.* Insertional mutagenesis identifies drivers of a novel oncogenic pathway in invasive lobular breast carcinoma. *Nat. Genet.* (2017). doi:10.1038/ng.3905
2. Kedziora, K. M. *et al.* Rapid Remodeling of Invadosomes by Gi-coupled Receptors: DISSECTING THE ROLE OF Rho GTPases. *J. Biol. Chem.* **291**, 4323–4333 (2016).
3. Zhou, H. & Kramer, R. H. Integrin engagement differentially modulates epithelial cell motility by RhoA/ROCK and PAK1. *J. Biol. Chem.* **280**, 10624–10635 (2005).
4. Bustos, R. I., Forget, M.-A., Settleman, J. E. & Hansen, S. H. Coordination of Rho and Rac GTPase function via p190B RhoGAP. *Curr. Biol. CB* **18**, 1606–1611 (2008).

5. Rosenfeldt, H., Castellone, M. D., Randazzo, P. A. & Gutkind, J. S. Rac inhibits thrombin-induced Rho activation: evidence of a Pak-dependent GTPase crosstalk. *J. Mol. Signal.* **1**, 8 (2006).
6. Alberts, A. S., Qin, H., Carr, H. S. & Frost, J. A. PAK1 negatively regulates the activity of the Rho exchange factor NET1. *J. Biol. Chem.* **280**, 12152–12161 (2005).
7. Guilluy, C., Garcia-Mata, R. & Burridge, K. Rho protein crosstalk: another social network? *Trends Cell Biol.* **21**, 718–726 (2011).

Reviewers' comments:

Reviewer #1 (Remarks to the Author):

The authors have improved the manuscript but I still find some points that need to be addressed in order for this work to be accepted for publication:

1. After demonstrating in this new version of ms that you do need some actomyosin to survive (as high conc of ROCK inhibitor or blebbistatin indeed killed the cells), I believe the general tone on the paper and the title should be changed from

Actomyosin relaxation enables tumor formation upon loss of E-cadherin expression in the mammary gland

To

A critical balance in Actomyosin levels enables tumor formation upon loss of E-cadherin expression in the mammary gland

This title reflects better the authors' observations

In the same line when in the intro the mention "Actomyosin is a critical barrier in ILC development", again "Actomyosin levels are critical in ILC development" better reflects the conclusions

2. Figure 2c,2d authors state "we did not observe amoeboid migration"- nevertheless the cells they describe are

- Highly motile
- Have high levels of RhoA
- have high levels of MLC activity
- and are blebbing
- not dying

This is the very definition of an amoeboid moving cell, so what are the other parameters (that are not shown in the manuscript) that made them decide they are not amoeboid? This is very unclear to this reviewer. From the data shown one cannot conclude that this is not amoeboid migration.

3. Figure 3g, authors should show levels of phospho-MLC to show cells do retain some actomyosin; and -as such- they should also shown it in figure 4c to prove that ROCK inhibitor used at different concentrations has an effect on phosphoMLC that is dose dependent and that way one can see the optimal levels of actomyosin that allow growth. Same in figure 5e, levels of Pmlc need to be shown

Reviewer #2 (Remarks to the Author):

I was supportive of this manuscript at the first review.

I had one major comment i.e. that the authors had not provided any data demonstrating that cells expressing a truncated-MYPT1 showed enhanced survival on collagen plates. In the revised manuscript, the authors have provided data (Fig. 6j,k) demonstrating that ectopic expression of tMYPT1 (but not tMYPT1delPPP1) indeed enhancing growth on collagen. In addition the authors have corrected minor comments.

I will not comment on the response to the other reviewer comments but I note that the authors have provided new data and substantially revised the manuscript

I am supportive of publication in Nature Communications
Reviewer #3 (Remarks to the Author):

The authors have addressed the main concerns and I recommend publication of this interesting work.

Reviewer #1 (Remarks to the Author):

The authors have improved the manuscript but I still find some points that need to be addressed in order for this work to be accepted for publication:

1. After demonstrating in this new version of ms that you do need some actomyosin to survive (as high conc of ROCK inhibitor or blebbistatin indeed killed the cells), I believe the general tone on the paper and the title should be changed from “*Actomyosin relaxation enables tumor formation upon loss of E-cadherin expression in the mammary gland*” to “*A critical balance in Actomyosin levels enables tumor formation upon loss of E-cadherin expression in the mammary gland*”. This title reflects better the authors' observations

Reply: We agree with the reviewer that balanced actomyosin contractility is required for tumor formation. Taking this into account we decided to change the title to: Rebalancing of actomyosin contractility enables mammary tumor formation upon loss of E-cadherin.

In the same line when in the intro the mention “Actomyosin is a critical barrier in ILC development”, again “Actomyosin levels are critical in ILC development” better reflects the conclusions

Reply: We agree with the reviewer and have changed the sentence according to the reviewer's suggestion (marked line 55-56 in the revised manuscript).

2. Figure 2c,2d authors state “we did not observe amoeboid migration”- nevertheless the cells they describe are

- Highly motile
- Have high levels of RhoA
- have high levels of MLC activity
- and are blebbing
- not dying

This is the very definition of an amoeboid moving cell, so what are the other parameters (that are not shown in the manuscript) that made them decide they are not amoeboid? This is very unclear to this reviewer. From the data shown one cannot conclude that this is not amoeboid migration.

Reply: We cannot exclude nor confirm that the observed motility is amoeboid migration since the cells tumble too rapidly to determine a defined form of cell movement. During the time of imaging we did not observe any cells using membrane blebs for directed movement. To highlight this more specifically we changed line 103-105 to: “Membrane blebbing is often seen in amoeboid migration¹⁶ and apoptosis¹⁷. However, we could not observe a defined form of cell movement or cell death during the time of imaging.”

3. Figure 3g, authors should show levels of phospho-MLC to show cells do retain some actomyosin; and -as such- they should also shown it in figure 4c to prove that ROCK inhibitor used at different concentrations has an effect on phosphoMLC that is dose dependent and that way one can see the optimal levels of actomyosin that allow growth. Same in figure 5e, levels of Pmlc need to be shown

Reply: We agree with the reviewer that E-cadherin-deficient MMECs need to retain a certain amount of actomyosin contractility to enable tumor formation. We have performed Western blots with E-cadherin-deficient MMECs exposed to a concentration range of H1152 during adhesion. The results which can be found in supplementary figure 4 of the revised manuscript show a dose dependent reduction in MLC phosphorylation. It is evident that during adhesion optimal amounts ROCK inhibition only partially inhibits MLC phosphorylation. The phospho MLC IF data in Figure 5c already show that even under expression of t-MYPT1 in E-cadherin deficient MMECs phosphorylation of MLC is not completely lost. We added a sentence to highlight this (line 203-204 in the revised manuscript).

Reviewer #2 (Remarks to the Author):

I was supportive of this manuscript at the first review.

I had one major comment i.e. that the authors had not provided any data demonstrating that cells expressing a truncated-MYPT1 showed enhanced survival on collagen plates. In the revised manuscript, the authors have provided data (Fig. 6j,k) demonstrating that ectopic expression of tMYPT1 (but not tMYPT1delPP1) indeed enhancing growth on collagen. In addition the authors have corrected minor comments.

I will not comment on the response to the other reviewer comments but I note that the authors have provided new data and substantially revised the manuscript

I am supportive of publication in Nature Communications

Reviewer #3 (Remarks to the Author):

The authors have addressed the main concerns and I recommend publication of this interesting work.

Reply: We thank the reviewers 2 and 3 for taking the time to read our revised manuscript and are glad we were able to address the remarks of the reviewers. We believe that the manuscript has been significantly improved due to their constructive feedback.